# Machine learning and deep learning based streamflow prediction in a hilly catchment for future scenarios using CMIP6-GCMs data

Dharmaveer Singh[1&5]*, Manu Vardhan[2], Rakesh Sahu[3], Debrupa Chatterjee[1], Pankaj Chauhan[4], Shiyin Liu[5]

1. Symbiosis Institute of Geo-informatics, Symbiosis International (Deemed University), Pune-411004 (India).
2. Computer Science and Engineering Department, National Institute of Technology Raipur, Raipur-492010 (India)
3. Computer Science and Engineering Department, Chandigarh University, Mohali- 140413 (India)
4. Wadia Institute of Himalayan Geology, Dehradun-248001 (India)
5. Institute of International Rivers and Eco-security, Yunnan University, Kunming - 650091 (China)

*Correspondence to: veermnnit@gmail.com and shiyin.liu@ynu.edu.cn

**Abstract**

The alteration in river flow patterns, particularly those that originate in the Himalayas, has been caused by the increased temperature and rainfall variability brought on by climate change. Due to the impending intensification of extreme climate events, as predicted by the Intergovernmental Panel on Climate Change (IPCC) in its sixth assessment report, it is more essential than ever to predict changes in streamflow for future periods. Despite the fact that some research has utilised machine learning and deep learning based models to predict streamflow patterns in response to climate change, very few studies have been undertaken for a mountainous catchment, with the number of studies for the western Himalaya being minimal. This study investigates the capability of five different machine learning (ML) models and one deep learning (DL) model, namely the Gaussian Linear Regression Model (GLM), Gaussian Generalized Additive Model (GAM), Multivariate Adaptive Regression Splines (MARS), Artificial Neural Network (ANN), Random Forest (RF), and 1D-Convolutional Neural Network (1D-CNN), in streamflow prediction over the Sutlej River Basin in the western Himalaya during the periods 2041-2070 (2050s) and 2071-2100 (2080s). Bias corrected data downscaled at grid resolution of $0.25° \times 0.25°$ from six General Circulation Models (GCMs) of the Coupled Model Intercomparison Project Phase 6-GCMs framework under two greenhouse gas trajectories (SSP245 and SSP585) were used for this purpose. Four different rainfall scenarios ($R_0$, $R_1$, $R_2$, and $R_3$) were applied to the models trained with daily data (1979-2009) at Kasol (the outlet of the basin) in order to better understand how catchment size and the geo-hydro-morphological aspects of the basin affect runoff. The predictive power of each model was assessed using six statistical measures: the coefficient of determination ($R^2$), the ratio of the root mean square error to the standard deviation of the measured data (RSR), the mean absolute error (MAE), the Kling-Gupta efficiency (KGE), the Nash-Sutcliffe efficiency (NSE), and the percent bias (PBIAS). RF model with rainfall scenario $R_3$ which outperformed other models during the training ($R^2$=0.90; RSR=0.32; KGE=0.87; NSE=0.87; PBIAS=0.03) and testing ($R^2$=0.78; RSR=0.47; KGE=0.82; NSE=0.71; PBIAS=-0.31) period therefore was chosen to simulate streamflow in the Sutlej River in the 2050s and 2080s under the SSP245 and SSP585 scenarios. Bias correction was further applied to the projected daily streamflow in order to generate reliable times series of the discharge. The mean ensemble of model results show that the mean annual streamflow of the Sutlej River is expected to rise between 2050s and 2080s by 0.79 to 1.43% for SSP585 and by 0.87 to 1.10% for SSP245. In addition, streamflow will increase during the monsoon (9.70 to 11.41% and 11.64 to 12.70%) in the 2050s and 2080s under both emission scenarios, but it will decrease during the pre-monsoon (-10.36 to -6.12% and -10.0 to -9.13%) and post-monsoon (-1.23 to -0.22% and -5.59 to -2.83%), as well as during the winter (-21.87 to -21.52% and -21.87 to -21.11%). This variability in streamflow is highly correlated with the pattern of precipitation and temperature predicted by CMIP6-GCMs for future emission scenarios, as well as with physical processes operating within the catchment. Predicted declines in Sutlej River streamflow over the pre-monsoon (April to June) and winter (December to March) seasons might have a significant impact on agriculture downstream of the river, which is already having problems due to water restrictions at this time of year. The present study will therefore assist in strategy planning for ensuring the sustainable use of water resources downstream by acquiring a knowledge of the nature and causes of unpredictable streamflow patterns.

**Keywords**: Machine learning models; 1D-CNN; streamflow; climate change; CMIP6-GCMs; western Himalaya

## 1  Introduction

Human-induced global warming has altered patterns of the rainfall worldwide (Goswami et al., 2006; Trenberth, 2011), and also increased risks of extreme events such as the droughts and floods (Easterling et al., 2000; Trenberth et al., 2015; Otto et al., 2017). It has impacted hydrology of many river basins globally, including variation in streamflow (Gerten et al., 2008; Nepal and Shrestha, 2015; Singh et al. 2015a; Ali et al., 2018; Lutz et al., 2019; Singh et al., 2022). A study of long-term (1948-2004) streamflow (discharge) data of 200 largest rivers of the globe showed considerable change in their annual discharge, however, results were statistically significant only for 64 rivers (Dai et al., 2009). Out of which 45 were marked with decreasing trends and the remaining 19 showed increasing trends in their annual discharge. Similar decreasing and increasing trends in discharge of the rivers were reported also at regional scale: Asia (Kundzewicz et al., 2009; Krysanova et al., 2015), Europe (Stahl et al., 2010; Stahl and Tallaksen, 2012) and America (Pasquini and Depetris, 2007). Moreover, it has been established that the effects of rainfall variation and extreme events on annual discharge are likely strong compared with other drivers (Kundzewicz et al., 2009; Miller et al., 2012; Van der Wiel et al., 2019). Zhao et al. (2021) examined how precipitation, evapotranspiration, and timing of snowmelt impacted runoff in the Kaidu River Basin of China. They discovered that as global warming increased, the timing of snowmelt became less significant while the influence of precipitation increased comparatively. A projected rise of ~2°C to 5°C in mean annual global temperature by 2100 under higher greenhouse gas emission scenarios as predicted from the General Circulation Models (GCMs) (Gao et al., 2017) will considerably affect the rainfall pattern (intensity and amount) and may alter hydrological cycles (Okai and Kanae 2006; Haddeland et al., 2014). This would subsequently impact availability of water resources and present challenges for their management since a rise in the demand of water is also predicted (Lutz et al., 2019). Therefore, it is indispensable to know the underlying hydrological dynamics occurring within a basin in context of climate change for effective management and sustainable use of the water resources.

The underlying hydrological processes controlling rainfall-runoff generation in a basin can be understood with the use of a hydrological model which is based on complex mathematical equations and theoretical laws governing physical processes in the basin (Kirchner, 2006; Singh et al., 2019). It simulates/or predicts response of the basin to climatological forcings such as the rainfall (Sood and Smakhtin, 2015) and generates synthetic time series of hydrological data that can be used by water managers and scientists for varied applications ranging from water budgeting and partitioning (Conan et al., 2003; Schreiner-McGraw and Ajami, 2020) to inundation mapping and modelling (Mahto et al., 2022). A hydrological model is supposed not only to have a good predictive power but also the ability of capturing relationships among the forcing factors and catchment response so that an accurate estimate of rainfall-runoff could be made (Shortridge et al., 2016). However, until now, there is no hydrological model that can simulate basin-behaviour universally well against all the hydrological challenges inflicted from climate change and human-interventions (Yang et al., 2019). As a result, many hydrological models have been devised considering functioning and robustness of models in explaining underlying complexity in quantifying basin-scale response to small-scale spatial complexity of physical processes (Shortridge et al., 2016; Herath et al., 2021). Broadly, these can be grouped into two categories: physical or process-based models and empirical or data-driven models (Yang et al., 2019; Kabir et al., 2020).

The latter category of models uses a mathematical relationship established between runoff and affecting factors
in the basin for deriving the runoff (Adnan et al., 2019).

It is purported that the data-driven model despite of inherited limitations over physical interpretability of
processes has outperformed the physical models in terms of prediction accuracy in many hydrological
applications (Shortridge et al., 2016; Adnan et al., 2019; Kabir et al., 2020; Herath et al., 2021). Also, they are
preferred over the physical models for rainfall-runoff modelling/or streamflow prediction modelling due to
limited requirements of data as inputs, where data limitation is the major challenge (Beven, 2011). These models
in past were heavily criticised on the ground of being incompetent to model the non-linear behaviour of
streamflow (Yang et al., 2019). But recent developments in computational intelligence, in the areas of machine
learning (ML) and deep learning (DL) in particular, have greatly expanded the capabilities of empirical
modelling (Adnan et al., 2020; Fu et al., 2020; Rahimzad et al., 2021; Ghobadi and Kang, 2022). This resulted
in the development of many non-linear models such as the Artificial Neural Network (ANN), Random Forest
(RF), Support Vector Regression (SVR) and Long Short-Term Memory (LSTM) models, which can capture and
model non-stationarity of the rainfall-runoff relationships (Yaseen et al., 2015; Shortridge et al., 2016; Adnan et
al., 2019; Yang et al., 2019; Xiang et al., 2020). Yang et al. (2019) applied three machine learning models
namely ANN, SVR, and RF to predict monthly streamflow over the Qingliu River basin in China under
changing environmental conditions between 1989 and 2010, and compared their results with the six process-
based hydrological models. They concluded that the ML model performed better than the process-based model
not just in terms of prediction accuracy, but also in terms of flexibility when it came to including other runoff
effect factors into the model. Similar outcomes for Lake Tana and the adjacent rivers in Ethiopia were also
reported by Shortridge et al. (2016), where ML models demonstrated noticeably lower streamflow prediction
errors than the physical models developed for the region. However, they inferred that linear machine learning
models, such as the Multivariate Adaptive Regression Splines (MARS) and Generalized Additive Model
(GAM), were sensitive to extreme climate events, so the degree of uncertainty in their predictions needed to be
carefully considered.

The limitations of such data-driven models can be overcome by adopting more advanced ML and DL models
(Xiang et al., 2020). Rasouli et al. (2012) compared the performance of the Multi-Linear Regression (MLR)
model with the Bayesian Neural Network (BNN), SVR, and Gaussian process (GP) in terms of daily streamflow
prediction for the Stave River, a mountainous basin, in British Columbia, and found that the BNN model
performed better than others. According to Hussain and Khan (2020), supervised learning model RF
outperformed Multilayer Perceptron (MLP) and SVR in terms of accuracy while predicting monthly streamflow
for the Hunza river in Pakistan by 33.6% and 17.85%, respectively. Recently, Deep Neural Network (DNN),
Convolutional Neural Network (CNN) and LSTM models, which are based on deep learning, have seen a surge
in the number of streamflow prediction applications due to their abilities to handle complex stochastic datasets
and abstracting the internal physical mechanism (Fu et al., 2020; Ghobadi and Kang, 2022). Based on statistical
performance evaluation criteria, Rahimzad et al. (2021) found that the LSTM outperformed the LR, SVM, and
Multilayer Perceptron (MLP) models in daily streamflow prediction over the Kentuky River basin in the USA.
However, Van et al. (2020) showed that CNN outperformed LSTM in streamflow modelling in the Vietnamese

Mekong Delta by a small margin. Comparing data-driven models to a given problem yield a range of results for distinct geographical and climatic conditions (Hagen et al., 2021. Adnana et al. (2020) examined the predictive accuracy of Optimally Pruned Extreme Learning Machine (OP-ELM), Least Square Support Vector Machine (LSSVM), MARS, and Model Tree (M5Tree) models in order to estimate monthly streamflow in the Swat River Basin (Hindukush Himalaya), Pakistan. They came to the conclusion that the LSSVM and MARS are the most effective at forecasting streamflow. In contrast, Hussain et al. (2020) discovered that ELM outperformed 1-D-CNN while forecasting streamflow on three time scales i.e., daily, weekly and monthly in the Gilgit River, Pakistan. This suggests that it is challenging to find a data-driven model that is effective across all application domains and scales (Yaseen et al., 2015; Fu et al., 2020).

The use of machine learning and deep learning based models for streamflow simulations within catchments is generally limited to observable periods and resulting forecasts (Eng and Wolock, 2022). There are very limited studies worldwide where these models were applied in predicting long-term streamflow for future periods in context of climate change (Das and Nanduri, 2018; Thapa et al., 2021; Adib and Harun, 2022). This can be attributed to the challenges associated with data assimilation brought on by the use of coarse resolution scenario data obtained from General Circulation Models (GCMs), which limits their direct application in regional impact assessment (Hagen et al., 2021; Adib and Harun, 2022). Das and Nanduri (2018) integrated Relevance Vector Machine (RVM) and SVM models with Coupled Model Intercomparison Project Phase (CMIP5)-GCMs to project monthly monsoon streamflow across the Wainganga basin (India) for monsoon season. Adib and Harun (2022) studied variations in the monthly streamflow pattern of the Kurau River (Malaysia) from 2021 to 2080 by coupling ML models (RF and SVR) with Coupled Model Intercomparison Project Phase (CMIP6)-GCMs. Despite of the significance potentials of the ML and DL models in streamflow prediction, relevant studies assessing the application of these models for streamflow prediction under future scenarios over the mountainous basins are limited due to non-availability of long-term data (Xenarios et al., 2019; Adnana et al., 2020). Thapa et al. (2021) used a combination of the LSTM model and the CMIP5-GCMs scenarios to estimate streamflow patterns in the Langtang basin of the Central Himalayas. Their analyses revealed a notable increase in streamflow as a result of the predicted increase in precipitation. The projections from Coupled Model Intercomparison Project Phase 3 (CMIP3)-GCMs and CMIP5-GCMs inherit limitations in simulating extreme precipitation (Kim et al., 2020), which are the principal drivers for the runoff generation in the catchment. This causes large uncertainty in streamflow predictions (Wang et al., 2021). Uncertainty in streamflow prediction can be minimised by using scenarios from the CMIP6-GCMs which are likely to be more realistic than previous generations, i.e., CMIP3-GCMs and CMIP5-GCMs, given their significant improvement in simulating rainfall and temperature for historical records (Chen et al., 2020; Gusain et al., 2020; Kim et al., 2020). Therefore, projected changes in streamflow patterns derived from CMIP6-GCMs scenarios would give a better understanding of the catchment's future hydrological regime than previous ones. To the authors' knowledge, no work has been published over a mountainous basin that integrates ML/DL models with CMIP6-GCMs scenarios to predict changes in streamflow patterns for future periods. Hence, it is important to test whether machine learning approaches can be effectively used over a mountainous river basin to predict streamflow using hydro-meteorological variables and CMIP6-GCMs scenarios as the input data.

With a catchment area of 56874km$^2$ (up to Bhakara Dam), the Sutlej also pronounced as 'Satluj' is an important
river in the western Himalayas and runs through diverse climatic zones. The flow in the upper and middle
catchment is primarily impacted by glacier/snow melt induced by seasonal temperature shift and preceding
winter precipitation, while the lower section of the catchment area is mostly regulated by rainfall both in the
winter and during the monsoon season (Singh and Jain, 2002; Archer, 2003; Miller et al., 2012). Based on data
from the period 1986–1996, Singh and Jain (2002) estimated the mean yearly contribution of snow/glacier melt
and rainfall to the Sutlej River as being 59% and 41%, respectively. However, the discharge in the river peaks is
directly related to the peak in rainfall during the monsoon (Lutz et al., 2014). Recent studies on this basin has
raised concerns about the implications of climatic changes on streamflow since a warming climate has brought
changes in the amount and spatial-temporal distribution of precipitation (Singh et al., 2014; Singh et al., 2015b).
Previous research has only used process-based hydrological models and scenarios from CMIP3-GCMs and
CMIP5-GCMs to date when examining the effects of climate change (past and future) on streamflow patterns in
the region (Singh and Jain, 2002; Singh et al., 2015a; Ali et al., 2018; Shukla et al., 2021), which leaves a gap in
the use of machine and deep learning models and scenarios from the latest CMIP6-GCMs.This study very first
time examines the potential of five ML models and one deep learning model namely, Gaussian Linear
Regression Model (GLM), Gaussian Generalized Additive Model (GAM), MARS, ANN, RF and 1D-CNN in
streamflow prediction over the middle Sutlej River Basin (rainfall dominated zone) in western Himalaya using
different Shared Socio-economic Pathways (SSPs) scenarios from CMIP6-GCMs. The pattern of variations in
the Sutlej River's monthly, seasonal, and annual streamflow are assessed for the future periods 2041-2070
(2050s) and 2071-2100 (2080s) with respect to the reference period of 1979-2009 under SSP245 and SSP585. .
The findings of the study will help to develop a better plan for the operation of hydroelectric power projects and
water resources management in the catchment.
**2   Study Area**
The selected study area is a sub-catchment within the Satluj basin (Figure 1), with an area of 2457 km$^2$.
Topographically, it is very rugged (0-80°) and is dominated mostly by forests (56.20%), grassland (26.4%),
agricultural lands (17.1%), and glaciers and snow covers (0.3%) (Singh et al., 2015a). The presence of mountain
barriers in the sub-basin's north, large variation in altitudes (500–5000 m) and the aspect all contribute to the
region's diverse climate. It varies from hot and moist tropical climate in lower valleys to cool temperate climate
at about 2000 m, and tends towards alpine as the altitude increases beyond 2000 m. The mean annual discharge
(averaged over the period of 1979-2009) of the river gauged at Kasol was 12469.43 m$^3$/s. There is large inter-
diurnal and monthly variation in pattern of the river discharge. The minimum and maximum daily discharge
recorded at Kasol was 64.30 m$^3$/s and 2891m$^3$/s, respectively. The early months of year, i.e., starting from
January up to March are characterised by low stream flow. After this a continuous and rapid rise in flow occurs,
being the maximum in the month of July (~22-23%). Then, it again starts decreasing and flow becomes the
minimum in the month of December (2-3%). The details of the sub-catchment are summarised in Table 1.
Figure 1: The location of the sub-catchment within Sutlej River Basin. The three hydro-meteorological stations
(Kasol, Sunni and Rampur) from which this study employed observed data for the years 1979 to 2009 are also
shown.

The sub-basin is bestowed with the large hydropower potential. There are three major hydroelectric power projects: Sunni Dam Project of 1080 MW, Rampur Hydroelectric Power Project (RHEP) of 412 MW, and Nathpa Jhakari Hydro-electric Power Project (NJHEP) of 1500 MW. The sub-basin is climatologically sensitive and, at present, facing the challenges created due to climate change and human's interventions (Singh et al., 2015b and 2015c). Change in future climate will alter patterns of flow in river and further could affect water resources and hydroelectric power production (Singh et al., 2014).

Table 1: Characteristics of the study catchment over the evaluation period of 1979–2009.

## 3 Description of the Data and Methods

The methodology involved in predicting streamflow for the period 2041-2100 in the Sutlej River include: 3.1) collection of hydro-meteorological data, 3.2) selection of machine and deep learning models, 3.3) performance evaluation of the developed models, and 3.4) bias correction in streamflow projection. These are described in details under following sub-headings:

### 3.1 Hydro-meteorological data

The daily rainfall, temperature ($T_{max}$ and $T_{min}$), relative humidity, solar radiation, wind speed and discharge data used to study performance of the different machine and deep learning models on streamflow modelling were collected for 31 years i.e. 1979-2009. Rainfall, temperature and discharge data were obtained from the Bhakara Beas Management Board (BBMB), while relative humidity, solar radiation and wind data were extracted from the Global Weather Data (http://globalweather.tamu.edu/). These data were collected for three hydro-meteorological stations namely, Kasol, Sunni and Rampur (Fig.1).

The downscaled outputs from the CMIP6-GCMs, the latest generation of climate models, were used for streamflow prediction in future (2050s and 2080s). This framework of CMIP6-GCMs was run to simulate future climate under four Shared Socio-economic Pathways Scenarios (SSPs), which are designed to explain potential future greenhouse gas emissions under various global socioeconomic shifts that would occur by 2100 (Riahi et al., 2017; Karan et., 2022). Even by using downscaled outputs, however, regional climate change projections inherit biases from the GCM boundary conditions (Jose and Dwarakish, 2022), which were corrected in the dataset detailed in Mishra et al. (2020) for South Asia. They used Empirical Quantile Mapping (EQM) method for removing bias in the downscaled data. This dataset provides bias-corrected downscaled climate change projections for 13 CMIP6-GCMs and four GHG emission scenarios (SSP126, SSP245, SSP370, and SSP585), the latter are briefly summarised in Riahi et al. (2017). Climate projections from CMIP6-GCMs that have been generated under the SSP245 and SSP585 scenarios were used in this study. SSP245, a medium scenario represents the average pathway of future greenhouse gas emissions with radiative forcing of 4.5 W/m² by the year 2100, while SSP585 is the upper limit of the range of scenarios scenario with radiative forcing of 8.5 W/m² by the end of this century (O'Neill et al., 2016). The data are available at a daily time-scale and horizontal spatial resolution of 0.25°×0.25°. Seven grids of the downscaled CMIP6-GCMs data cover the study area. The temperature ($T_{max}$ and $T_{min}$) data were adjusted for topographical bias by separating the study area into a number of homogenous elevation bands spaced by at an interval of 1000m, and applying a temperature laps rate of

6.5°C/1000m within each grid. A Digital Elevation Model (DEM) of 30 m spatial resolution derived from
CartoSat-1 stereo data (www.bhuvan.nrsc.gov.in) was used for this purpose. The values of rainfall and
temperature at each grid were then averaged over the catchment using the Thiessen polygon method in order to
provide daily rainfall data integrated at the catchment scale for assessing changes in the future climate with
respect to the observed period i.e., 1979-2009.

Further, ranking of CMIP6-GCMs was done to find out the most appropriate models that can generate most
likely plausible scenarios of future climate in the catchment and ultimately being employed in streamflow
projection. Taylor diagram (Taylor, 2001), a robust graphical plot, is widely used to rank GCMs due to its
effectiveness in determining the relative strengths of the competing models and in evaluating overall
performance as a model evolves (Abbasian et al., 2019; Ghimire et al., 2021). It integrates three statistical
metrics, degree of correlation (r), centered root-mean-square error (CRMSE) and ratio of spatial standard
deviation (SD). Combining these metrics allows determining the degree of pattern correspondence and
explaining how exactly a model represents the observed climate (Taylor, 2001). Therefore, performance of 13
CMIP6-GCMs in modelling climatic variables (rainfall, $T_{max}$ and $T_{min}$) in the Sutlej sub-basin was compared to
the observed data (1979-2009) using Taylor diagram (Fig. 2a-c). The models were then ranked as a result of this
comparison. High positive correlation (r=0.84 to 0.96) and low CRMSE (<3°C) error were found in all 13
CMIP6-GCMs for temperature ($T_{max}$ and $T_{min}$) (Fig. 2b-c). Additionally, it was found that models' standard
deviations, which ranged from 5.60 to 6.03°C for $T_{max}$ and 6.34 to 6.63°C for $T_{min}$, were close to the SD of the
observed data (6.01°C and 6.07 °C). These results imply that all CMIP6-GCMs may be able to predict most
likely future temperature over the catchment.
Figure 2: Taylor diagram showing comparative skills of 13CMIP6-GCMs in simulating climatic variables
(rainfall, $T_{max}$ and $T_{min}$) over the Sutlej sub-basin during reference period (1979-2009). The degree of correlation
coefficient (r) between observed and CMIP6-GCMs, centered root-mean-square error (CRMSE) and departure
of the models' standard deviation (SD) from the observed data (dashed black arc line) are shown in Fig. 2a for
rainfall, Fig. 2b for $T_{max}$ and Fig. 2c for $T_{min}$. The units of SD for rainfall and temperature is in cm and °C,
respectively.
However, not all CMIP6-GCMs showed the high degree of similarity in predicting rainfall; in fact, two
(CanESM5 and NorESM2-LR) of the 13 models revealed a negative correlation (Fig. 2a). In the pool of 13
CMIP6-GCMs, only six models showed relatively higher correlation (r≥0.56), smaller CRMSE (<12 cm) errors,
and a high similarity to the standard deviation of the observed data (13.2 cm). They were: 1) Earth Consortium-
Earth 3 Veg Model (EC-Earth-Veg) , 2)  Russian Institute for Numerical Mathematics Climate Model Version
4.8 (INM-CM4-8), 3) Russian Institute for Numerical Mathematics Climate Model Version 5.0 (INM-CM5-0),
4) Max Planck Institute for Meteorology Earth System Model version 1.2 with higher resolution (MPI-ESM1-2-
HR) , 5) Max Planck Institute for Meteorology Earth System Model version 1.2 with lower resolution (MPI-
ESM1-2-LR) and 6) Norwegian Earth System Model Version 2 with Medium Resolution (NorESM2-MR).
Further, within these models, the highest and lowest correlations between observed and simulated rainfall were
found for the INM-CM4-8 (r=0.69) and NorESM2-MR (r=0.56), respectively. These six CMIP6-GCMs were
finally selected to examine future patterns in streamflow for the periods 2050s and 2080s in the Sutlej River
Basin as they had also shown high performance in simulating temperatures (r=0.90 to 0.96).

## 3.2 Selection of machine and deep learning models for streamflow modelling

In this study, five machine and one deep learning models namely GLM, GAM, MARS, ANN, RF and one dimensional Convolution Neural Network (1D-CNN) were selected and their performances in predicting streamflow in Sutlej River were compared. These are regression based models which capture relationship between the predictors (dependent variables) and predictand (independent variables) and provide value of the output variables (Adnan et al., 2019; Kabir et al., 2020). The models were trained with daily observed data recorded during 1979-2009 at Kasol (the gauging site) as well as simulated historical projections of CMIP6-GCMs. The climatic projections of the grid corresponding to Kasol station were taken into consideration as the input from the CMIP6-GCMs. However, prior to building the models, all of the data were normalized using standard normalization techniques to get features on a common scale. Further, the entire data set was split into training and testing datasets since a cross-validation method was adopted in this study. The training dataset (80%) was used for fitting the models whereas testing dataset was used for checking model accuracy (20%). Under the cross-validation method, the process was repeated until every part of the allocated data was used in testing (Kabir et al., 2020). Six different program codes were written in python language for ANN, GAM, GLM, MARS, RF and 1D-CNN simulations. Out of these six selected models, GLM, GAM and MARS are linear models whereas other three i.e. ANN, RF and 1D-CNN are non-linear in nature (Shortridge et al., 2016; Yang et al., 2019; Herath et al., 2021). Additionally, excluding GLM all of the remaining models are based on non-parametric regression approach where functional relationship between predictor and predictand are not predetermined but can be adjusted to capture unusual or unexpected features of the data (Shortridge et al., 2016). A detailed description of these models can be found elsewhere (Shortridge et al., 2016; Adnan, 2019; Yang et al., 2019; Kabir et al., 2020; Ghimire et al., 2021; Herath et al., 2021; Shu et al.,2021).

Since the 1D-CNN model is based on weight sharing, it needs less training parameters than other models (Kiranyaz et al., 2021). It has mainly three layer, convolution layer, pooling layer and fully connected layer. The primary job of the convolution layer is to nonlinearly map input data into a set of feature maps, or series of feature vectors. When working as a visual cortical perceptron, filter kernels are convoluted with the input data of their receptive fields. The convolution results with biases are then passed on to the activation function to create feature maps. The pooling layer, which comes after each convolution layer, primarily serves to reduce the dimension of feature maps and maintain the invariance of characteristic scale. The fully connected layer uses a completely connected single layer perceptron to combine the feature maps that were acquired by the prior convolution and pooling layers in order to build a higher level feature (Kiranyaz et al., 2021). In this study, one convolution layer with 64 filters, a kernel of size 2, and a ReLU activation function was being employed. This was followed by max pooling layer with pool size =2, and the faltterm layer. After that two fully connected layer applied with ReLU activation function and linear activation function, respectively. However, for optimization, the adaptive moment estimation (Adam) algorithm was applied (Ghimire et al., 2021; Shu et al.,2021). Six variables namely rainfall, $T_{max}$, $T_{min}$, relative humidity, solar radiation and wind speed were used as the inputs for developing the models. Additionally, these models were simulated under four rainfall scenarios: rainfall on the same day ($R_0$), rainfall lagged by one day ($R_1$) and rainfall lagged by two days ($R_2$) and rainfall lagged by three days ($R_3$) to understand control of catchment size and geo-hydro-morphological characteristics

of the basin in generating runoff. While, remaining meteorological parameters were held constant during the
processes.

### 3.3 Model performance evaluation

It has been found that overfitting in a model may lead to large errors in out-of-sample predictions (Hastie et al.,
2009). Therefore, it has been evaded by establishing model parameters for GLM, GAM, MARS, ANN and RF
through automated hyperparameter tuning methods. 500 bootstrap resamples of the training data set were
generated for each parameter value to be assessed. Table 2 presents the information on the specific parameters
evaluated for each model.
Table 2: The information on hyper parameters used for estimating model parameters.
The accuracy with which the simulated flow matches the observed flow during the training (calibration) and
testing (validation) phases determines whether a hydrological model is appropriate for a given application
(Refsgaard, 1997). Several methods, including quantitative statistics and graphical methods, has been developed
in the past for assessing the accuracy of model predictions (Legates and McCabe, 1999). Moriasi et al. (2007)
grouped these methods into three categories namely, standard regression, dimensionless, and error index,
depending on how well each method explains the relationship between observed and simulated values, compares
the relative performance of models, and quantifies the deviation in the units of the data of interest. Moreover, it
has been established from previous studies that a single metric is inadequate to evaluate a model's performance,
hence multiple metrics should be used (Adnan et al., 2020). Therefore, in this study, prediction accuracy of
different models was compared using six statistical measures out of which one was standard regression
(coefficient of determination ($R^2$)), two of which were dimensionless (Kling-Gupta efficiency (KGE) and Nash-
Sutcliffe efficiency (NSE)), and the remaining three were being error index (ratio of the root mean square error
to the standard deviation of the measured data (RSR)), the mean absolute error (MAE) and the percent bias
(PBIAS)). These metrics are defined below by the equations (2–7):

$$R^2 \text{ (Van Liew et al., 2003)} = \left( \frac{\sum_{i=1}^{n}(Q_i - \bar{Q})(P_i - \bar{P})}{\sqrt{\sum_{i=1}^{n}(Q_i - \bar{Q})^2} \times \sqrt{(P_i - \bar{P})^2}} \right) \quad \text{(range: 0 to 1)} \tag{1}$$


$$KGE \text{ (Gupta et al., 2009)} = 1 - \sqrt{(r-1)^2 + \left(\frac{\sigma_p}{\sigma_{0b}} - 1\right)^2 + \left(\frac{P_i}{Q_i} - 1\right)^2} \quad \text{(range: 0 to 1)} \tag{2}$$


$$NSE \text{ (Nash and Sutcliffe, 1970)} = 1 - \left[\frac{\sum_{i=1}^{n}(Q_i - P_i)^2}{\sum_{i=1}^{n}(Q_i - \bar{Q})^2}\right] \quad \text{(range: } -\infty \text{ to 1)} \tag{3}$$


$$RSR \text{ (Singh et al., 2004)} = \frac{\sqrt{\sum_{i=1}^{n}(Q_i - P_i)^2}}{\sigma_{ob}} \quad \text{(range: 0 to } \infty\text{)} \tag{4}$$


$$MAE \text{ (Adnan et al., 2020)} = \frac{\sum_{i=1}^{n}|P_i - O_i|}{n} \quad \text{(range: 0 to } \infty\text{)} \tag{5}$$


$$PBIAS \text{ (Gupta et al., 1999)} = \left[\frac{\sum_{i=1}^{n}(Q_i - P_i)}{\sum_{i=1}^{n}(Q_i)}\right] \times 100 \quad \text{(range: } -100 \text{ to } 100\%\text{)} \tag{6}$$


where $P_i$ are the predicted values and $Q_i$ are the observed values, $n$ accounts for the number of samples, $Q^-$ represents the mean of observed data, and $P^-$ is the mean of predicted data. However, $r$ is the Pearson's correlation coefficient whereas $\sigma_{ob}$ and $\sigma_p$ refers to the standard deviation of observed and predicted values, respectively.

$R^2$ evaluates the percentage of the variation in the measured data that can be explained by the model, whereas NSE estimates the relative size of the residual variance in relation to the variance in the measured data (Nash and Sutcliffe, 1970; Van Liew et al.,2003). According to Mazrooei et al. (2021), NSE is sensitive to extreme flows; as a result, KGE is also used to evaluate a model's performance while considering extreme flows into account (Adib and Harun, 2022). Other metrics, like RSR, MAE, and PBIAS, shed light on the overall inaccuracies in the projected flow relative to the observed. The value of $R^2$, KGE and NSE should all be 1 in an ideal model, whereas RSR and MAE and PBIAS values should be 0 (Nash and Sutcliffe,1970; Van Liew et al.,2003; Gupta et al.,2009; Adnan et al., 2020). Moriasi et al. (2007) developed a guideline for interpreting the results of these metrics and ranking for the hydrological models based on a thorough review of the available literature. They found that a model can be classified as very good, good, satisfactory, or unsatisfactory if its NSE value is between 0.75 and 1, 0.65 to 0.75, 0.50 to 0.65, or less than 0.50, respectively. Similarly, $R^2$ values between 0.6 to 0.7 are considered satisfactory, 0.85 to 1 are very good and below 0.5 are unsatisfactory (Van Liew et al., 2003). However, for RSR, numbers above 0.7 are considered to be poor, whereas values between 0 and 0.5 are considered to be in the very good range. Thus, the lower is the RSR value, the better is the model. This is also true for PBIAS and MAE where lower values are favourable. According to Moriasi et al. (2007), PBIAS values of less than ±10% are considered to be highly acceptable, whilst values of more than ±25% are considered to be unsatisfactory. The negative number indicates that the model has overestimated its bias, whereas the positive value indicates that the model has underestimated its bias (Gupta et al., 1999).

### 3.4 Bias correction

Uncertainty in streamflow prediction may be caused by the GCMs' shortcomings (e.g., coarse spatial resolution, simplified physics and thermodynamic processes, numerical methods, or poor knowledge of climate system dynamics) in accurately replicating natural climate variability (Sperna Weiland et al., 2010). As a result, its quantification and correction are critical for generating a future time series of streamflow that is reliable and recommended to devising water resource management plans in the catchment. This study used the bias correction method proposed in Hawkins et al. (2013) to correct uncertainty (bias) between observed and CMIP6-GCMs predicted streamflow. The mathematical expression for this formulae is given below:

$$Q_{bc} = \bar{Q}_{ob} + \frac{\sigma_{ob}}{\sigma_p}\left(Q_{future} - \bar{Q}_{p)}\right) \tag{7}$$

where, $Q_{bc}$ and $Q_{future}$ is the bias corrected and raw daily discharge for future simulation, respectively. $\bar{Q}_{ob}$ and $\bar{Q}_{p)}$ is the mean discharge of observed and historical simulation for reference period (1979-2009), respectively. $\sigma_o$ and $\sigma_p$ is the standard deviation in observed and historical simulation for reference period, respectively. This method captures variability in both observation and GCMs simulations Hawkins et al. (2013), which is the interest of this study.

## 4   Results

### 4.1   Streamflow simulation and evaluation of model performance

The simulation (1979-2009) results generated under different rainfall scenarios ($R_0$, $R_1$, $R_2$ and $R_3$) on daily time scale for all six models (GLM, GAM, MARS, ANN, RF and 1D-CNN) during training and testing is shown in Fig. 3 and Fig. 4, respectively. The model performed slightly better during training than testing periods. $R^2$, NSE and KGE values across models ranged from 0.69 to 0.90, 0.52 to 0.87, 0.69 to 0.91 and from 0.69 to 0.81, 0.49 to 0.74 and 0.68 to 0.82 during training and testing, respectively. Likewise, it was found that RSR, MAE and PBIAS varied from 0.31 to 0.55, from 71.95 to 123.25 $m^3$/s and -2.11 to +4.31% during training, as well as from 0.56 to 0.46, from 123.06 to 106.64 $m^3$/s and -3.74 to +2.21% during testing, respectively. Non-linear models (ANN,1D-CNN and RF) outperformed linear models (GAM and GLM) in runoff prediction under all rainfall scenarios ($R_0$, $R_1$, $R_2$, and $R_3$),with the exception of MARS, which produced results that were more or less comparable to those of the ANN model. Figures 3–4 show that both models (RF and 1D-CNN) satisfy the performance requirements outlined by Moriasi et al. (2007) as the best models, but RF slightly outperformed CNN in terms of error index. $R^2$, NSE, KGE, RSR, and MAE and PBIAS values for the RF model during the training ranged from 0.88 to 0.90, 0.85 to 0.87, 0.86 to 0.87, 0.32 to 0.34, 71.95 to 77.49 $m^3$/s and +0.03 to +0.13%, respectively. For the 1D-CNN, however, it varied from 0.87 to 0.89, 0.85 to 0.87, 0.90 to 0.91, 0.34 to 0.35, 80.29 to 83.14 $m^3$/s, and -1.25 to +0.13%. Similar pattern with slightly lower values were revealed during testing for the both models. This implies that RF can effectively capture non-linear interactions and can provide insights about actual watershed functions (Shortridge et al., 2016). On the other hand, GLM showed the poorest results. $R^2$, NSE, KGE, RSR, MAE, and PBIAS values for the GLM model during the training varied from 0.69 to 0.71, 0.52 to 0.56, 0.71 to 0.72, 0.54 to 0.55, 134.80 to 140.56 $m^3$/s, and +2.63 to +2.73%, respectively. During testing, they varied between 0.69 and 0.71, 0.49 and 0.54, 0.68 and 0.70, 0.54 and 0.56, 134.35 and 141.26 $m^3$/s, +1 and +1.31%, respectively. Furthermore, it was observed that the models with rainfall scenario $R_3$ had revealed reasonably better results in comparison to $R_0$, $R_1$ and $R_2$ scenarios, indicating delayed contribution of rainfall-runoff to the river.

Figure 3: Evaluation of the model (ANN, 1D-CNN, GAM, GLM, MARS and RF) performance in simulating streamflow under rainfall scenarios $R_0$ (Fig.3a), $R_1$ (Fig. 3b), $R_2$ (Fig.3c) and (Fig. 3d) $R_3$ at Kasol during training phase using six statistical metrics ($R^2$, KGE, NSE, RSR, MAE and PBIAS).

Figure 4: Evaluation of the model (ANN, 1D-CNN, GAM, GLM, MARS and RF) performance in simulating streamflow under rainfall scenarios $R_0$ (Fig.4a), $R_1$ (Fig.4b), $R_2$ (Fig.4c) and (Fig.4d) $R_3$ at Kasol during testing phase using six statistical metrics ($R^2$, KGE, NSE, RSR, MAE and PBIAS).

Figure 5, 6, 7 and 8 shows comparison of observed and simulated streamflow under rainfall scenarios of $R_0$, $R_1$, $R_2$ and $R_3$ for all the models at Kasol, the outlet of the basin. As observed from the Figures (5-8), RF was able to follow the curve better compared to the other models. It is also deduced from the comparison of scatter plots wherein a relatively smaller deviation in the observed and estimated discharge of streamflow was found for the RF model. GLM performed the worst out of the six models with respect to the time variation graphs. A limitation faced by all the six models was the simulation of peak values. The models slightly underperformed at the prediction of higher values of streamflow. These findings led to the ultimate decision to use the RF model

with rainfall scenario $R_3$ to predict streamflow in the Sutlej River in the future (2050s and 2080s) under the
SSP245 and SSP585 scenarios.
Figure 5: Comparison of observed and simulated streamflow for all six models (ANN, 1D-CNN, GAM, GLM,
MARS and RF) under rainfall scenarios $R_0$
Figure 6: Comparison of observed and simulated streamflow for all five models (ANN, 1D-CNN, GAM, GLM,
MARS and RF) under rainfall scenarios $R_1$
Figure 7: Comparison of observed and simulated streamflow for all five models (ANN, 1D-CNN, GAM, GLM,
MARS and RF) under rainfall scenarios $R_2$
Figure 8: Comparison of observed and simulated streamflow for all five models (ANN, 1D-CNN, GAM, GLM,
MARS and RF) under rainfall scenarios $R_3$.
**4.2  Comparison of streamflow simulated with observed and CMIP6-GCMs data**
The uncertainty between observed and CMIP6-GCMs predicted streamflow during the reference period (1979-
2009) was investigated by comparing the streamflow simulated by RF model with observed and CMIP6-GCMs
data. A large difference in streamflow patterns was seen in the box-plot of observed and CMIP6-GCMs
simulated discharge (Fig. 9) derived for various months of the year, particularly from June through September
(monsoon season), when a pattern of intense daily rainfall was observed over the catchment. Additionally, it was
discovered through the analysis of probability exceedance curves generated using 10% of the time series' highest
flows that, despite the streamflow's in the two data sets being comparable throughout the pre-monsoon season
(Fig. 10c), they differ noticeably for high flows during the annual (Fig.10a) and monsoon season (Fig.10c).
Similar trends were seen in the comparison of the probability exceedance curves for low flows during the
monsoon season, although there was strong agreement for annual (Fig.10b) and pre-monsoon measurements
(Fig.10d). This may be due to the fact that orography has a considerable impact on regional Indian Summer
Monsoon (ISM) climate, making it challenging for climate models to predict daily monsoonal rainfall
accurately across the Himalaya (Turner and Annamalai, 2012; Niu et al., 2015; Choudhary et al., 2017). The
Regional Climate Model (RCM) based on CMIP5-GCMs was used by Sanjay et al. (2017) to study pattern of
change in precipitation and temperature over the HKH region. As a condition of the model's inability to
accurately represent complicated feedback mechanisms, the results revealed large uncertainty in the summer and
winter precipitation over the northwest Himalaya. This is also supported by the study of Kadel et al. (2018).
They evaluated the performance of 38 CMIP5-GCMs in simulating rainfall over the central Himalaya and came
to the conclusion that the majority of the models' studied performed poorly when it comes to reproducing the
spatial distribution of monsoonal rainfall. Although the most recent study by Gusain et al. (2020) in India
reported that ISM simulation using CMIP6-GCMs over CMIP5-GCMS had significantly improved, there are
discrepancies between the models and indicated uncertainty in predictions. Lalande et al. (2021) examined the
abilities of 26 CMIP6-GCMs to simulate the rate of precipitation across the Himalayan region and concluded
that the models consistently overestimated the rate of precipitation by 31% to 281%. Additionally, cold-bias in
temperature estimation was also reported. Therefore, bias correction as described in Section 3.4 was applied to
the projected streamflow for the future periods (2050s and 2080s) under all scenarios and for all six models in
order to provide accurate times series of the discharge.

Figure 9: Box-plot comparing observed and CMIP6-GCMs (mean ensemble of models) simulated streamflow for various months of the year, derived over the period of 1979–2009. The line inside the box denotes the median values of streamflow, while the upper and lower whiskers indicate the highest and minimum values, respectively.

Figure 10: Probability exceedance curves developed using 10% of the highest and lowest flows from the observed and CMIP6-GCMs (mean ensemble of models) over the time span of 1979–2009 for annual and seasonal (pre-monsoon and monsoon) flows.

**4.3 Projected change in rainfall and temperatures in 2050s and 2080s under SSP245 and SSP585**

Figure 11 shows how the catchment's mean monthly rainfall is expected to change under SSP245 and SSP585 in the 2050s and 2080s compared to the reference period (1979-2009). Within months and for the CMIP6-GCMs, a sizable shift in the rainfall pattern is seen. With the exception of March, June, and September, the mean ensemble of the models generally predicts a rise in rainfall throughout the year in the 2050s and 2080s under all scenarios. The models also show significant variation in the seasonal and yearly rainfall patterns expected for the catchment in the 2050s and 2080s under various emission scenarios. However, based on the mean ensemble of the models, it is predicted that seasonal (Fig. 12) and annual (Fig. 13a) rainfall will increase generally in the 2050s and 2080s under SSP245 and SSP585. Pre-monsoon, monsoon, post-monsoon, and winter rainfall in 2050s will increase by 8.75 to 8.85%, 10 to 20.80%, 85 to 91.91%, and 12.48 to 14.16%, respectively, under SSP245 and SSP585. However, under SSP245 and SSP585 in the 2080s, it will rise by 7.69 to 17.50%, 21.52 to 41.43%, 56.16 to 89.66%, and 22.48 to 12.43%, respectively. Under both scenarios in the 2050s and 2080s, pre-monsoon and post-monsoon will have the lowest and highest percentage increases in rainfall, respectively. The monsoon season, however, is anticipated to have the greatest rise in terms of quantity (~40-167mm). The predicted range for the increase in mean annual rainfall is 13.85 to 18.61% in the 2050s and 17.91% to 34.31% in the 2080s. It is observed that the predicted pattern of change in rainfall across the sub-basin under various SSPs is consistent in terms of the direction of change with other studies conducted over the Sutlej and Himalaya region. Lalande et al. (2021) reported an overall increase in mean annual precipitation over the Himalayan region based on 10 CMIP6-GCMs. According to their analysis, the mean ensemble of model precipitation is predicted to increase by 8.6% to 25.4% in 2081-2100 under SSP245 and SSP585. The same study also showed an increase in the region's winter (November to April) and ISM (June to September) rainfall. This contradicts past studies that showed a trend toward declining ISM rainfall after the 1950s (Sabin et al., 2020). They postulated that the region's higher winter rainfall would have been caused by the strengthening of the western disturbances; however, the intensification of the ISM is responsible for the region's enhanced summer rainfall.

Figure 11: Projected change in mean monthly rainfall in the sub-basin using different CMIP6-GCMs under SSP245 and SSP585 scenarios in the 2050s (Fig.11a and Fig.12b) and 2080s (Fig.12c and Fig.12d).

Figure 12: Projected change in mean seasonal rainfall in the sub-basin using different CMIP6-GCMs under SSP245 and SSP585 scenarios in the 2050s (Fig.12a and Fig.12c) and 2080s (Fig.12b and Fig.12d).

Figure 13: Projected change in mean annual rainfall (Fig.13a), $T_{max}$ (Fig.13b) and $T_{min}$ (Fig.13c) in the sub-basin using different CMIP6-GCMs under SSP245 and SSP585 scenarios in the 2050s and 2080s.

The analysis of the CMIP6-GCM projections leads to the conclusion that for all months and seasons in the
2050s and 2080s, maximum (excluding April and pre-monsoon in 2050s under SSP245) and minimum
temperatures will rise under both scenarios (Fig. 14 (a-d) and Fig.15 (a-d)). Similarly, increase in mean annual
$T_{min}$ and $T_{max}$ are also predicted in 2050s and 2080s under all scenarios (Fig.13b and 13c). The increase will be
relatively higher for the $T_{min}$ as compared to the $T_{max}$. This is also reported by Singh et al. (2015c). The increase
in rainfall and temperature is typically higher under SSP585 than SSP245 in both eras (2050s and 2080s), as
expected, due to a larger increase in radiative forcing brought on by increased greenhouse gas emissions.
Figure 14: Projected change in mean seasonal maximum temperature ($T_{max}$) in the sub-basin using different
CMIP6-GCMs under SSP245 and SSP585 scenarios in the 2050s (Fig.14a and Fig.14 c) and 2080s (Fig.14b and
Fig.14d).
Figure 15: Projected change in mean seasonal minimum temperature ($T_{min}$) in the sub-basin using different
CMIP6-GCMs under SSP245 and SSP585 scenarios in the 2050s (Fig.15a and Fig.15c) and 2080s (Fig.15b and
Fig.15d).

### 522 4.4 Assessment of change in streamflow in 2050s and 2080s under SSP245 and SSP585

The Sutlej River's mean monthly streamflow change as compared to the reference period's observed flow (1979-
2009) is shown in Fig. 16 under scenarios SSP245 and SSP585 for the future periods (2050s and 2080s).
According to both scenarios and all six models, the Sutlej River's streamflow will decrease between January (-
33.80 to -14.38%), February (-32.40 to -14.15%), March (-23.55 to -0.84%), November (-21.06 to -5.14%) and
December (-29.88 to -18.38%) in the 2050s and 2080s. Moreover, except for MPI-ESM-2HR and MPI-ESM1-
2-LR, which show an increase in streamflow in the 2080s under the higher emission scenario, all of the CMIP6-
GCMs indicate a decrease in the river's discharge in June (-20.24 to -0.57%) under SSP245 and SSP585 for both
the periods. Similarly, excluding EC-Earth-Veg (under SSP245 in 2050s) and INM-CM5-0 (under SSP245 in
250s and 2080s and under SSP585 in 2050s), all of the CMIP6-GCMs indicate a decrease in the river's
discharge in May (-25 to -2.85%) during the study period. In contrast, under SSP245 and SSP585 in the 2050s
and 2080s, all of the CMIP6-GCMs predict a rise in the river's discharge in April (20.24 to -0.57%; excluding
SSP585 in 2080s), August (16.84 to 5.28%), and September (55.27 to 4.35%). But no clear pattern of
streamflow change is seen for the remaining months (July and October) of the year, making results difficult to
generalise because projected decrease/or increase in streamflow over the months is inconsistent among models
under various emission scenarios in the 2050s and 2080s. The variations in climate variable projections caused
by differing spatial resolutions and parametrization levels in the climate models may be the cause of these
discrepancies in streamflow estimates (Sperna Weiland et al., 2010; Singh et al., 2015a). According to Murphy
et al. (2004), the average of an ensemble of GCMs cancels out the errors of each individual model, and as more
models are used, the ensemble uncertainty decreases. Therefore, in order to reduce uncertainty in projection of
streamflow related to individual CMIP6-GCMs, streamflow pattern of the Sutlej River was analysed also using
the mean ensemble of all six GCMs.
Figure 16: Predicted change in monthly streamflow pattern of the Sutlej River with respect to the reference
period (1979-2009) in 2050s (Fig.16a and Fig. 16b) and 2080s (Fig.16c and Fig. 16d) under SSP245 and
SSP585 for different CMIP6-GCMs.

The mean ensemble of the models predicts that the Sutlej River's mean monthly streamflow (excluding April) will decrease under both scenarios from November (-18.45 to -17.17%) to June (-10.90 to -8.06%) between 2050s and 2080s (Fig. 17). The river flow, which would have been expected to increase in April under both scenarios in 2050s, will also decline in 2080s for the higher emission scenarios (SPP585). The maximum and minimum streamflow declines are predicted to occur in the 2050s under SSP245 for the months of December (-24.25%) and May (-7.77%), respectively. In comparison to SPP245, the decline generally will be slightly higher under SSP585 in 2050s and, for the 2080s, the projected decrease in streamflow will not show much difference under both the scenarios. Opposite to this, the mean ensemble of the models predicts that the Sutlej River's flow will increase from July (5.50 to 5.91%) to October (3.01 to 11.42%) in the 2050s and 2080s under both the scenarios. The maximum and minimum streamflow increases are predicted to occur in the 2080s under SSP245 for the months of September (25.82%) and July (5.50%), respectively. In all scenarios, the increase will be slightly greater in the 2080s than it will be in the 2050s. When compared to SPP245, it will be higher for SSP585 in scenarios.

Figure17: Comparison of monthly observed (1979-2009) and projected discharge of the multi-model ensembles for period 2050s and 2080s under SSP245 and SSP585 scenarios.

The projected change in seasonal streamflow of the Sutlej River in 2050s and 2080s is shown in the Fig. 18. The 2050s and 2080s would see an increase in streamflow during the monsoon (4.46 to 16.14%) and a decrease during the pre-monsoon (-17.40 to -0.51%) and winter (-28.81 to -12.42%) for all six CMIP6-GCMs, with the exception of INM-CM5-0 in the 2050s under SSP245 and MPI-ESM-2HR and MPI-ESM1-2-LR in the 2080s under SPP585, which indicate an increase in streamflow during the pre-monsoon rather than a decrease. The predicted streamflow for the post-monsoon season, however, does not show a consistent pattern of change across time within the models under SSP245 and SSP585 scenarios. But there is high probability, based on the mean ensembles of models projections, that streamflow will also decline during the post-monsoon in 2050s (-1.23 to -0.22%) and 2080s (-5.59 to -2.83%) under all scenarios. Similarly, the predicted decline for pre-monsoon and winter will be between -10.36 and -6.12% and -21.87 and -21.52% under SSP245, and between -10.0 and -9.13% and -21.87 and -21.11% under SSP585, respectively. With the exception of winter, when there are no significant differences in the projected streamflow, the decline will be slightly larger in the 2080s than it would be in the 2050s in all scenarios. In addition, the results of the mean ensemble of the models indicate that the Sutlej River's flow will increase during the monsoon under both scenarios, from 9.70 to 11.41% in the 2050s and11.64 to 12.70% in the 2080s.

Figure 18: Predicted change in seasonal streamflow pattern of the Sutlej River with respect to the reference period (1979-2009) in 2050s (Fig. 18a and Fig. 18c) and 2080s (Fig. 18a and Fig. 18c) under SSP245 and SSP585 for different GCMs.

Similarly, Fig. 19 lists the projected change in mean annual streamflow for the Sutlej River in 2050s and 2080s with respect to the reference period (1979-2009) under different emission scenarios. Although the nature of the direction of change within models vary, the mean ensemble of the models reveals a persistent increasing pattern in streamflow for all scenarios in 2050s and 2080s. The Sutlej River's annual stream flow will rise between 2050 and 2080 by 0.79 to 1.43% for SSP585 and 0.87 to 1.10% for SSP245, according to the mean ensemble of the models. The rise is expected to be higher in 2080s as compared to 2050s under SSP585.

Figure 19: Predicted change in mean annual streamflow of the Sutlej River with respect to the reference period (1979-2009) in 2050s and 2080s under SSP245 and SSP585 for different GCMs.

## 5    Discussion

This study reveals an increase in the Sutlej River's mean annual and monsoonal streamflow in the 2050s and 2080s in contrast to earlier studies (Singh et al., 2014; Ali et al., 2018) that reported a reduction based on long-term investigation of station data over historical era. The pattern of rainfall and temperature predicted by CMIP6-GCMs for future periods under the SSP245 and SSP585 emission scenarios, as well as physical processes occurring within the basin, have contributed to this increase in the Sutlej River's streamflow. For instance, it is speculated that the projected increase in mean streamflow during the monsoon season under both scenarios in the 2050s and 2080 for all models is related to the projected percentage increase in rainfall amount over the catchment and the melting of glaciers brought on by the increased maximum and minimum temperatures. This increase in river streamflow and its propensity to raise silt load may have an impact on both the capacity of reservoirs and the hydropower potential of hydroelectric facilities situated in the sub-basin and downstream of it. On the other hand, despite the predicted increase in rainfall throughout the pre-monsoon, post-monsoon, and winter seasons, the anticipated decrease in streamflow of the Sutlej River during pre-monsoon, post-monsoon, and winter may be explained by the projected rise in temperatures, which may have led to increased evaporation from the surface. Similar conclusions were reached by Adib and Harun (2022) who studied the Kurau River in Malaysia and predicted a drop in streamflow during the months of January, April, and October despite receiving more rainfall. Moreover, during winter and post-monsoon, most of precipitation in upper part of the catchment occurs in form of snowfall which have minimal effect over runoff generation in the catchment. Additionally, the large increase in monsoonal streamflow predicted during study periods is what led to the projected increase in the Sutlej River's mean annual flow. Predicted decreases in Sutlej River streamflow over the pre-monsoon (April to June) and winter (December to March) seasons may have a significant impact on agriculture and hydropower generation downstream of the river, which is already struggling due to water shortages at this time of year. Ali et al. (2018) predicted that the hydroelectric production from the Nathpa Jhakri and Bhakra Nangal hydropower projects will decline during May to June in the future due to projected decline in the streamflow of the Sutlej River.

The projected streamflow patterns for the Sutlej River under SSP245 and SSP585 in 2050s and 2080s show similar tendencies, but with differing magnitudes, that have been found by past researchers using process-based hydrological models. For instance, Singh et al. (2015a) used the SWAT (Soil Water Assessment Tool) model, a semi-distributed hydrological model, to simulate streamflow for future periods using two CMIP3-GCMs models (CGCM3 and HadCM3), and they discovered that the Sutlej River's mean annual streamflow would increase in the range of 0.6 to 7.8% for the future periods (2050s and 2080s). Similar to this, using the Variable Infiltration Capacity (VIC) and SWAT models, respectively, Ali et al. (2018) and Shukla et al. (2021) estimated increases in the Sutlej River's mean annual streamflow for the 2050s and 2080s under RCP4.5 and RCP8.5. The study of Shukla et al. (2021) estimated that under RCP4.5 and RCP8.5, the mean streamflow of the river would increase by 14 and 21% (at Rampur), respectively, in the 2080s. The previous studies' observed substantially higher increase in projected streamflow may be attributable to the CMIP3-GCMs' and CMIP5-GCMs' overestimation

of monsoonal precipitation over the Himalayan region (Choudhary et al., 2017; Sanjay et al., 2017; Gusain et al., 2020; Lalande et al., 2021). Similar to this, the results of Singh et al. (2015a), Ali et al. (2018), and Shukla et al. (2021) corroborated the expected decrease in streamflow during pre-monsoon and winter as well as rise during monsoon. This suggests that the RF model can accurately predict runoff and analyse the effects of climate change while capturing the nonlinearity of a hilly catchment.

## 6    Conclusion

This study compared the performance of the five machine learning models (GLM, GAM, MARS, ANN, and RF) and one deep learning model (1D-CNN) which were further divided into linear (MARS, ANN, and RF) and non-linear (ANN, 1D-CNN, and RF) models, in simulating rainfall-runoff responses over the hilly Sutlej River Basin in order to determine the best model for predicting streamflow response to future climate change in the 2050s and 2080s under SSP245 and SSP585 using CMIP6-GCMs data. The important findings of the study are summarised below:

In general, non-linear models (ANN,1D-CNN and RF) outperformed linear models (GAM, GLM and MARS) in runoff prediction under all rainfall scenarios ($R_0$, $R_1$, $R_2$, and $R_3$). Among all the models, RF and 1D-CNN were identified as the best models as per the model evaluation criteria. However, RF outperformed CNN in terms of error index (MAE and PBIAS), and as a result, it was used to investigate impact of future climate change on the Sutlej River pattern in the 2050s and 2080s under SSP245 and SSP585 emission scenarios.

- The developed RF model slightly underperformed at the prediction of higher values of streamflow during training and testing. This implies that it is less effective in predicting flash floods that are caused by intense rainfall in the catchment. However, it was determined that the results produced by RF were comparable to process-based hydrological models for long-term change study in streamflow pattern.

- Significant variations in the streamflow pattern were observed throughout the periods of months, seasons, years, and for the CMIP6-GCMs. The differences in spatial resolution and parametrisation levels of CMIP6-GCMs, which caused a noticeable change in the projected amounts of temperature and precipitation during the study periods, may serve as an illustration of these variances in streamflow prediction. The Sutlej River's mean annual streamflow based on the mean ensemble of models is predicted to rise between the years 2050 and 2080 by 0.79 to 1.43% for SSP585 and by 0.87 to 1.10% for SSP245. Additionally, under both emission scenarios, streamflow will decrease during the pre- and post-monsoon (-1.23 to -0.22% and -5.59 to -2.83%), as well as during the winter (-21.87 to -21.52% and -21.87 to -21.11%), but increase during the monsoon (9.70 to 11.41% and 11.64 to 12.70%) in the 2050s and 2080s.

- The increase in the Sutlej River's streamflow (annual and monsoon) is due to both physical processes that occur within the basin and rainfall and temperature patterns that are predicted by CMIP6-GCMs for future time periods under the SSP245 and SSP585 emission scenarios. The projected rise in mean streamflow during the monsoon season is associated to both the projected percentage increase in

rainfall over the catchment and the melting of glaciers brought on by the increasing maximum and
minimum temperatures. On the other hand, the predicted increase in temperatures, which may have led
to increased evaporation from the surface, may be used to explain the anticipated reduction in
streamflow of the Sutlej River during pre-monsoon, post-monsoon, and winter.

•   Additionally, the projected changes in the mean annual and seasonal streamflow of the river are
consistent with earlier research done using process-based physical hydrological models. Thus, the
outcomes of the overall study indicate that the RF model is efficient for simulating streamflow in the
Himalayan catchment, and that water availability during monsoon will rise as a result of an increase in
catchment precipitation, which would eventually lead to an increased sediment load and affect
hydropower generation. However, predicted reduction in streamflow during pre-monsoon, post-
monsoon and winter will put stress on agriculture and hydropower generation downstream of the river,
which is already struggling due to water shortages at this time of year. The administrators of local
water resources and the government organizations in charge of maintaining reservoirs down river may
find these details on streamflow patterns to be of great use.

**Code Availability:** The codes developed for this study is made available to the readers on reasonable request.

**Data Availability:** The observed station data are confidential and authors do not have permission for sharing
the data.

**Author's Contribution**
DS and SL conceptualized the problems, supervised the entire research activity from its inception to the
completion, contributed in data collection, processing, interpretation and wrote the research paper. MV and RS
contributed in the development of model, generation of figures and analysis of data. PC and DC contributed in
the data analysis and interpretation.

**Statements and Declarations**
The authors declare that they have no known competing financial interests or personal relationships that could
have appeared to influence the work reported in this paper.

**Acknowledgement**
Authors acknowledge National Natural Science Foundation of China (NFSC Grant no. 42171129) for funding
this research work, and Bhakara Beas Management Board (BBM), India for the hydro-meteorological data used
in this study. We thank Ms. Pratibha Shrivastava, a fourth-year B.Tech computer science student from NIT
Raipur, for her help in building the model.

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

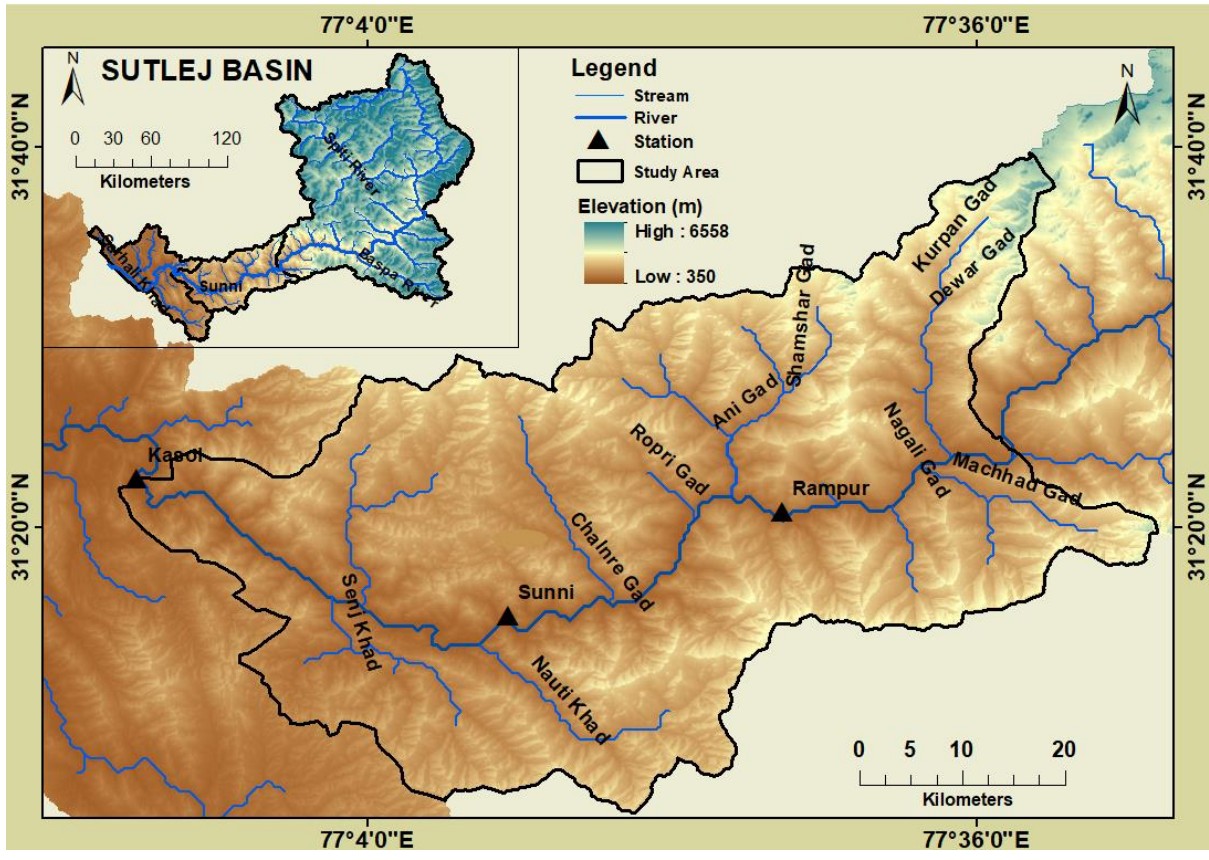


Figure 1: The location of the sub-catchment within Sutlej River Basin. The three hydro-meteorological stations (Kasol, Sunni and Rampur) from which this study employed observed data for the years 1979 to 2009 are also shown.


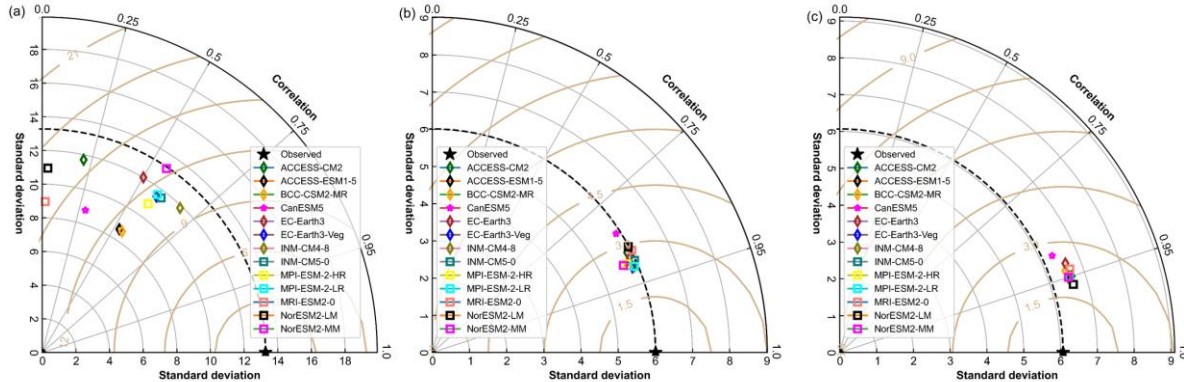

**Figure 2: Taylor diagram showing comparative skills of 13CMIP6-GCMs in simulating climatic variables (rainfall,**
**$T_{max}$ and $T_{min}$) over the Sutlej sub-basin during reference period (1979-2009). The degree of correlation coefficient (r)**
**between observed and CMIP6-GCMs, centered root-mean-square error (CRMSE) and departure of the models'**
**standard deviation (SD) from the observed data (dashed black arc line) are shown in Fig. 2a for rainfall, Fig. 2b for**
**$T_{max}$ and Fig. 2c for $T_{min}$. The units of SD for rainfall and temperature is in cm and °C, respectively.**

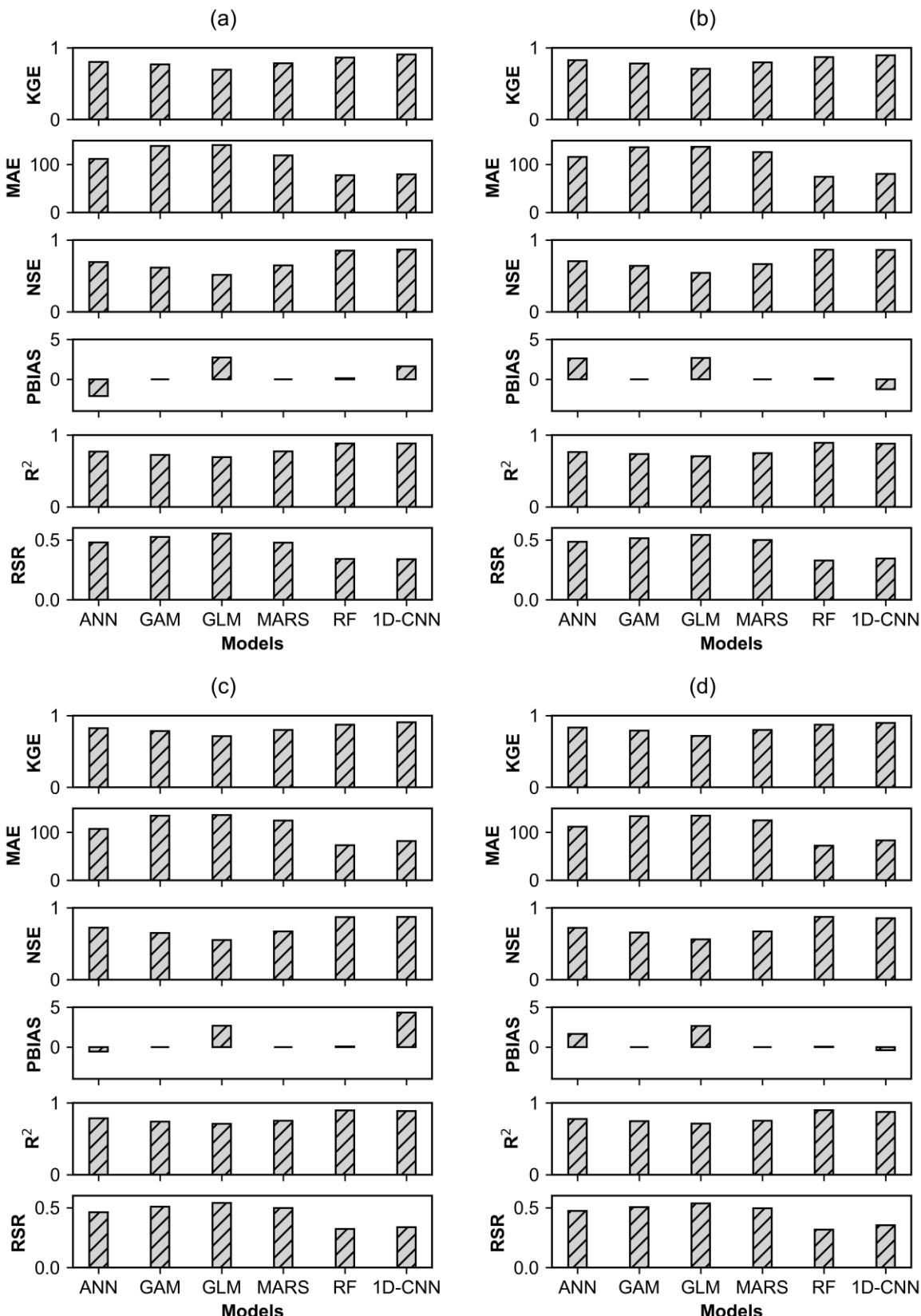

**Figure 3: Evaluation of the models (ANN, GAM, GLM, MARS, RF and 1D-CNN) performance in simulating**
**streamflow under rainfall scenarios $R_0$ (Fig.3a) , $R_1$ (Fig. 3b), $R_2$ (Fig.3c) and (Fig. 3d) $R_3$ at Kasol during training**
**phase using six statistical metrics ($R^2$, KGE, NSE, RSR, MAE and PBIAS).**

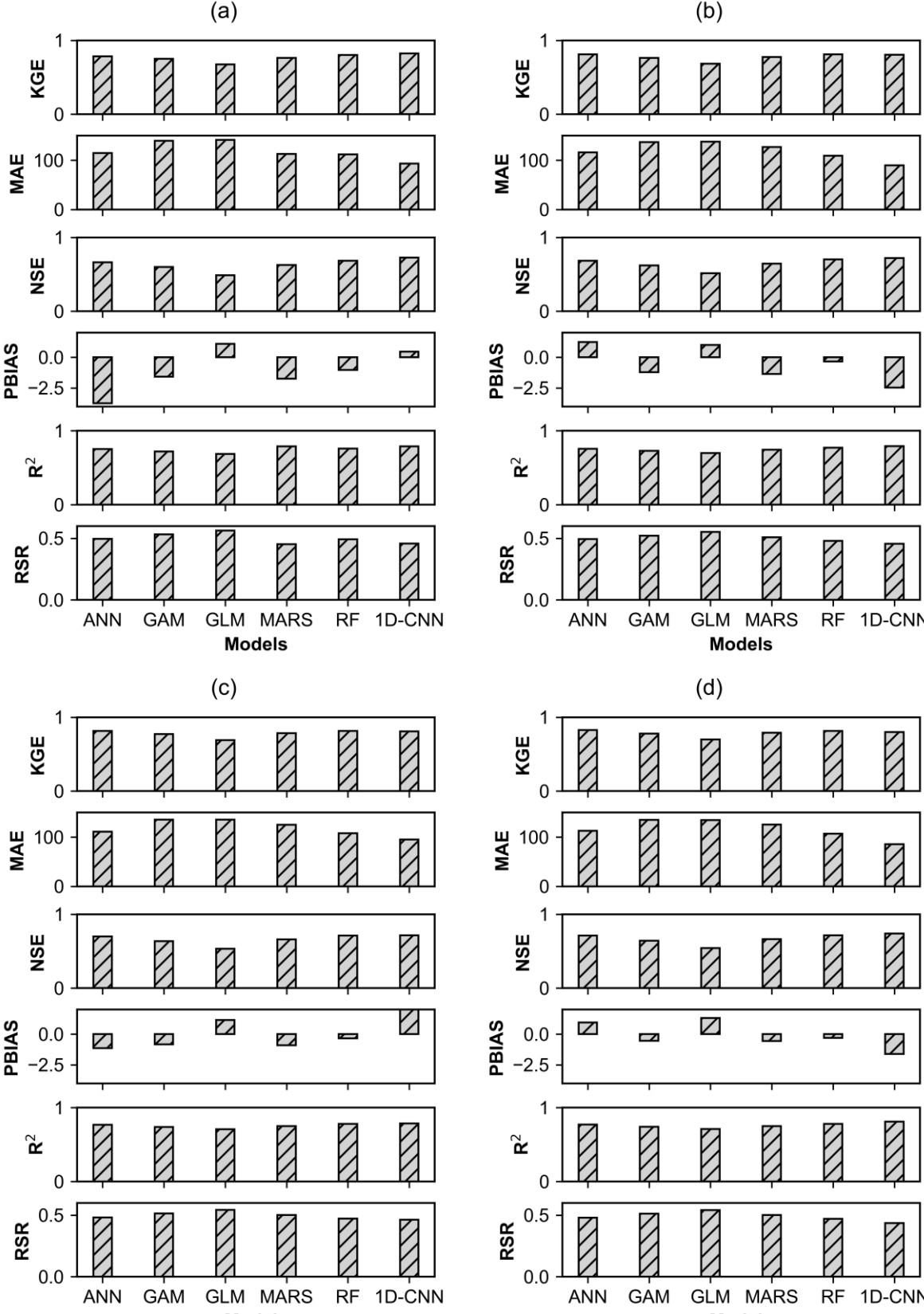

**Figure 4: Evaluation of the models (ANN, GAM, GLM, MARS, RF and 1D-CNN) performance in simulating streamflow under rainfall scenarios $R_0$ (Fig.4a), $R_1$ (Fig. 4b), $R_2$ (Fig.4c) and (Fig. 4d) $R_3$ at Kasol during testing phase using six statistical metrics ($R^2$, KGE, NSE, RSR, MAE and PBIAS).**


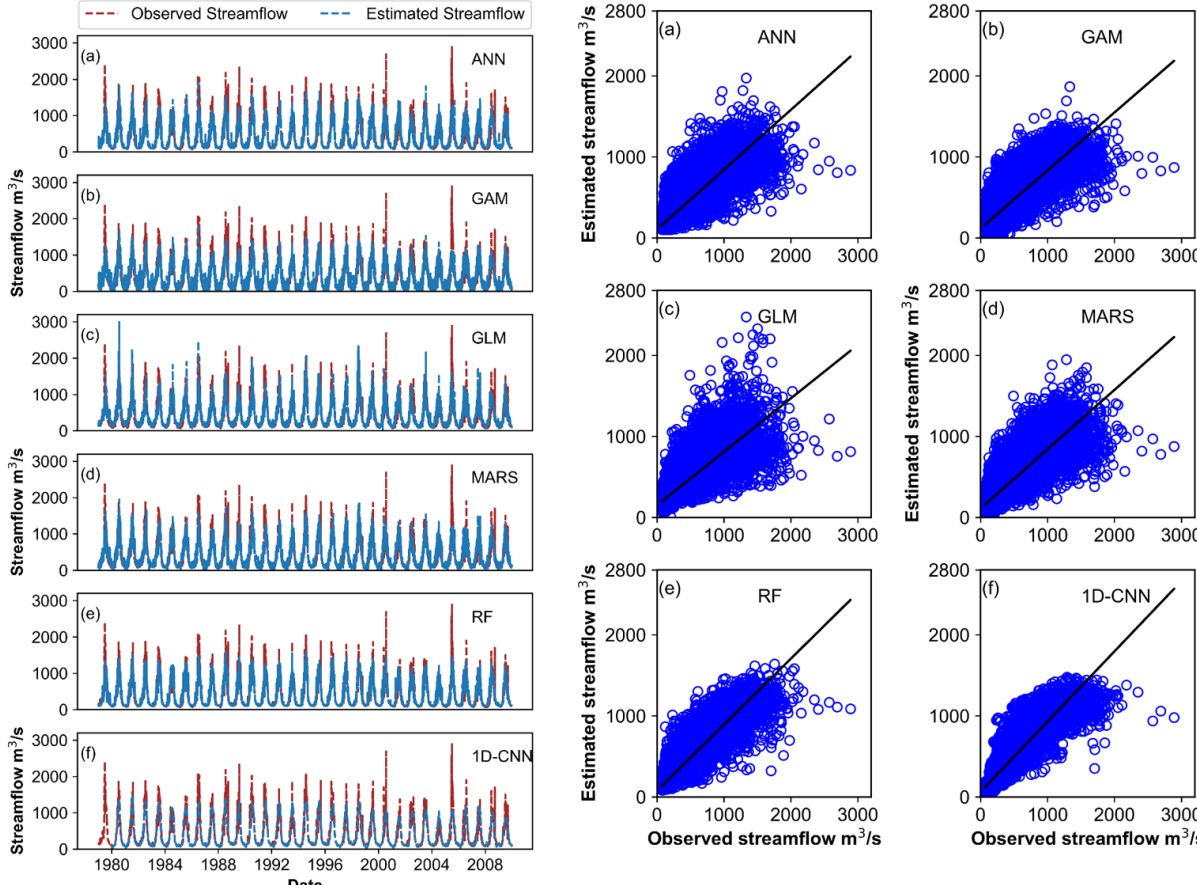


**Figure 5: Comparison of observed and simulated streamflow for all six models (ANN, GAM, GLM, MARS, RF and 1D-CNN) under rainfall scenarios R$_0$.**




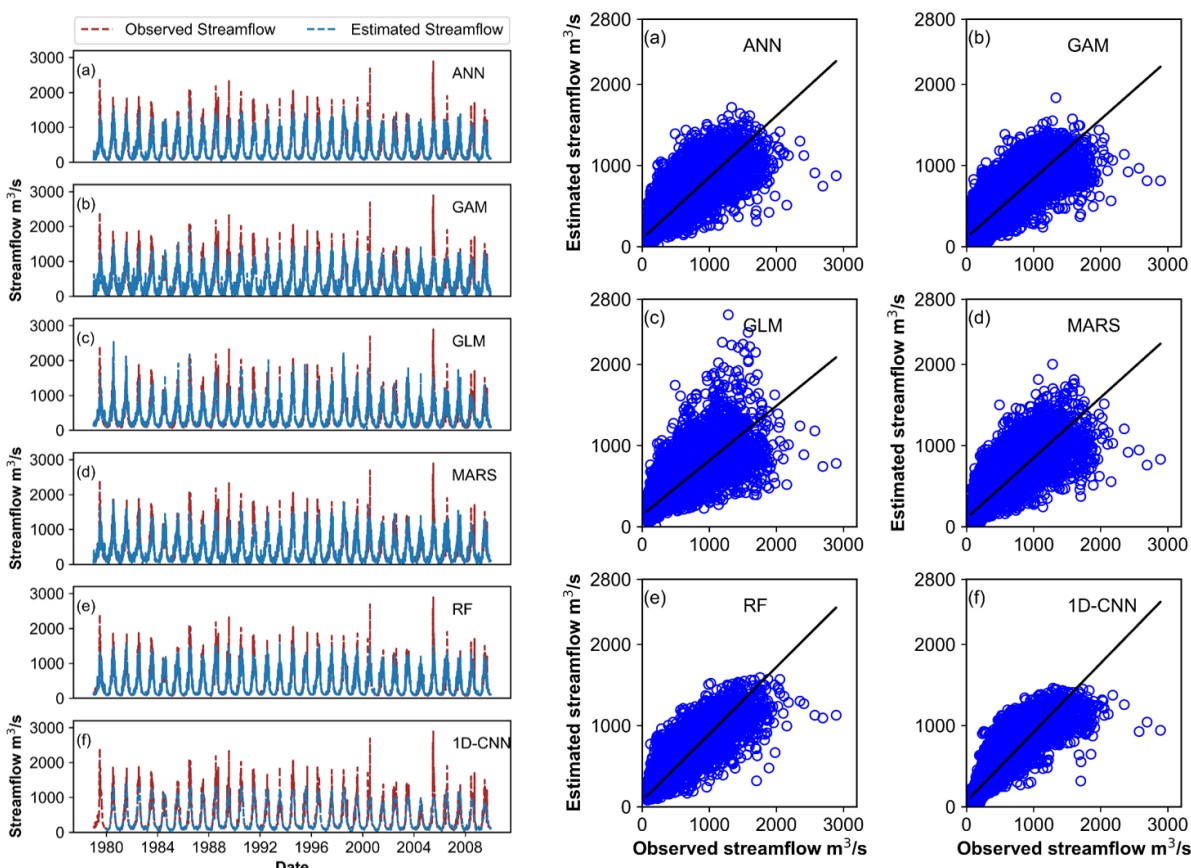


**Figure 6: Comparison of observed and simulated streamflow for all six models (ANN, GAM, GLM, MARS, RF and 1D-CNN) under rainfall scenarios R$_1$.**




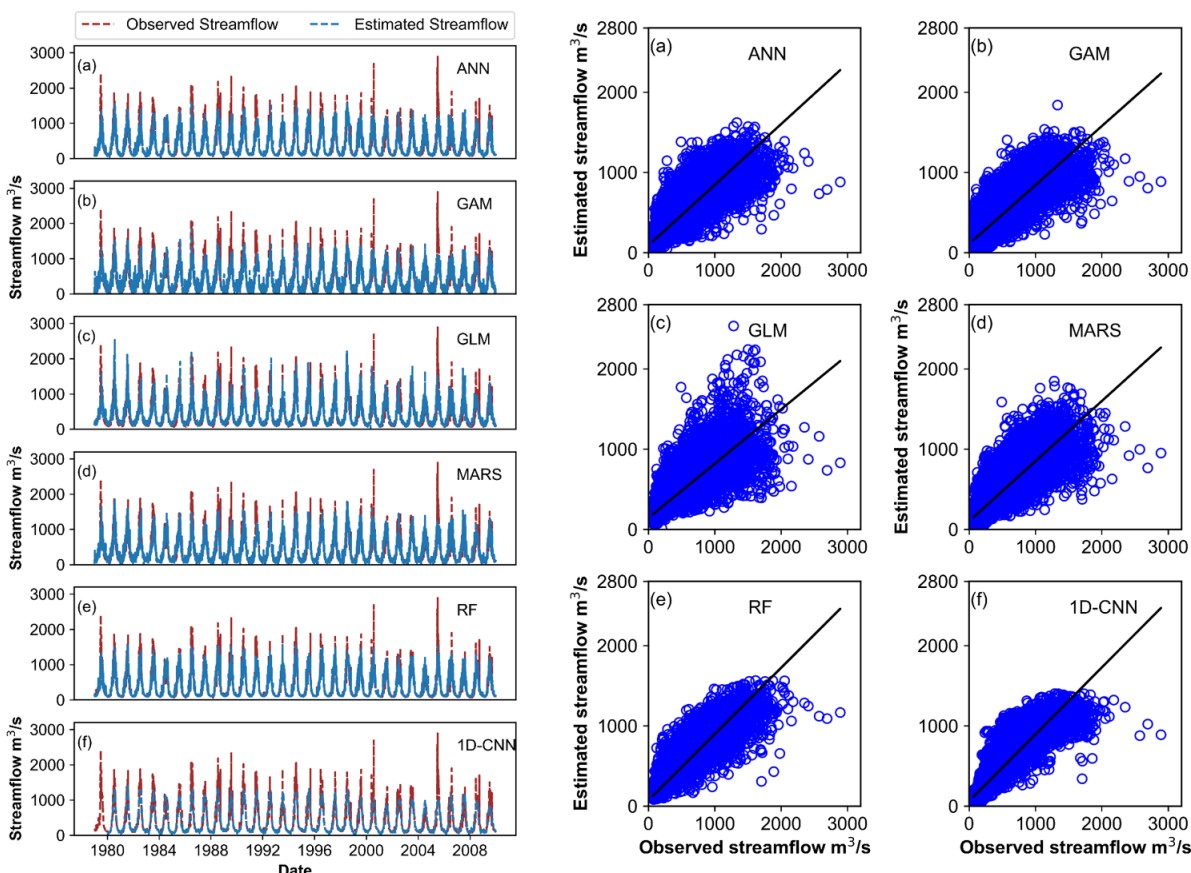


**Figure 7: Comparison of observed and simulated streamflow for all six models (ANN, GAM, GLM, MARS, RF and**
**1D-CNN) under rainfall scenarios R$_2$.**





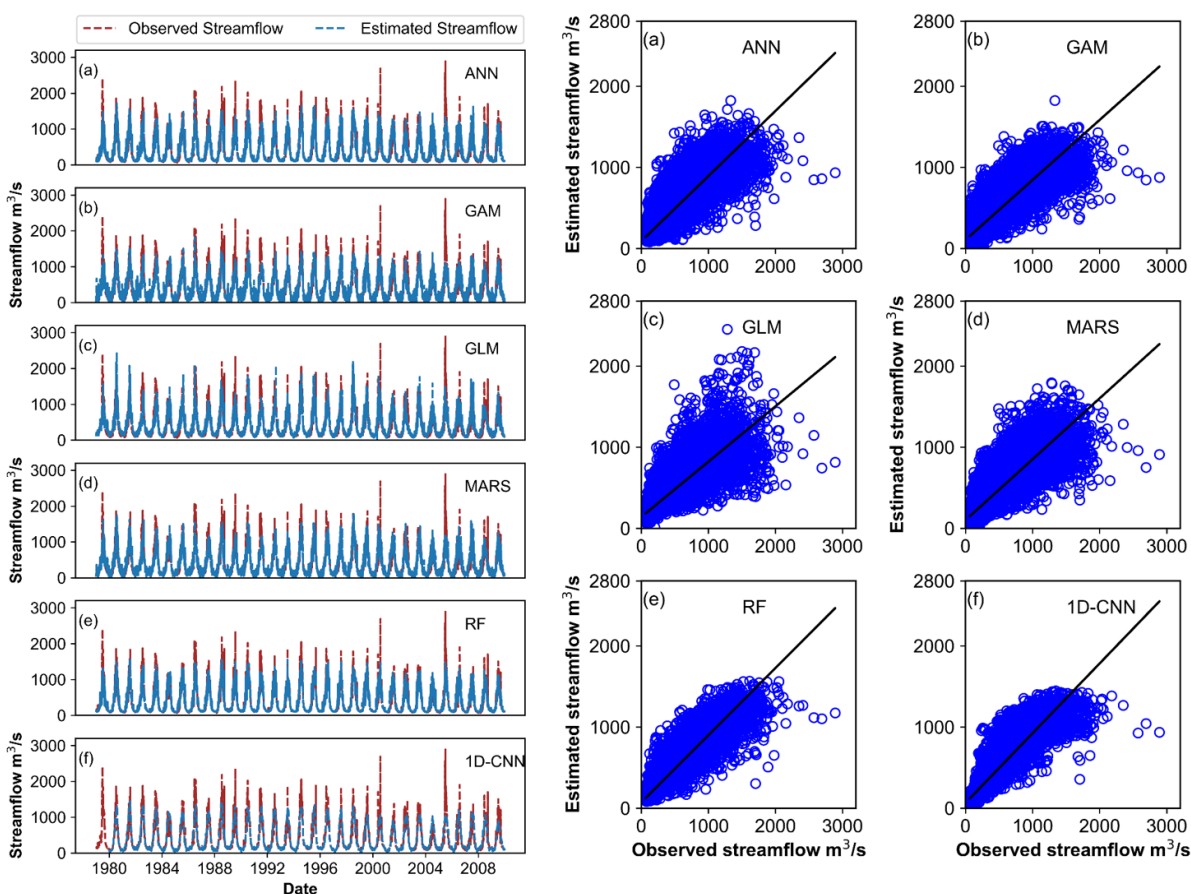


**Figure 8: Comparison of observed and simulated streamflow for all six models (ANN, GAM, GLM, MARS, RF and**
**1D-CNN) under rainfall scenarios R₃.**

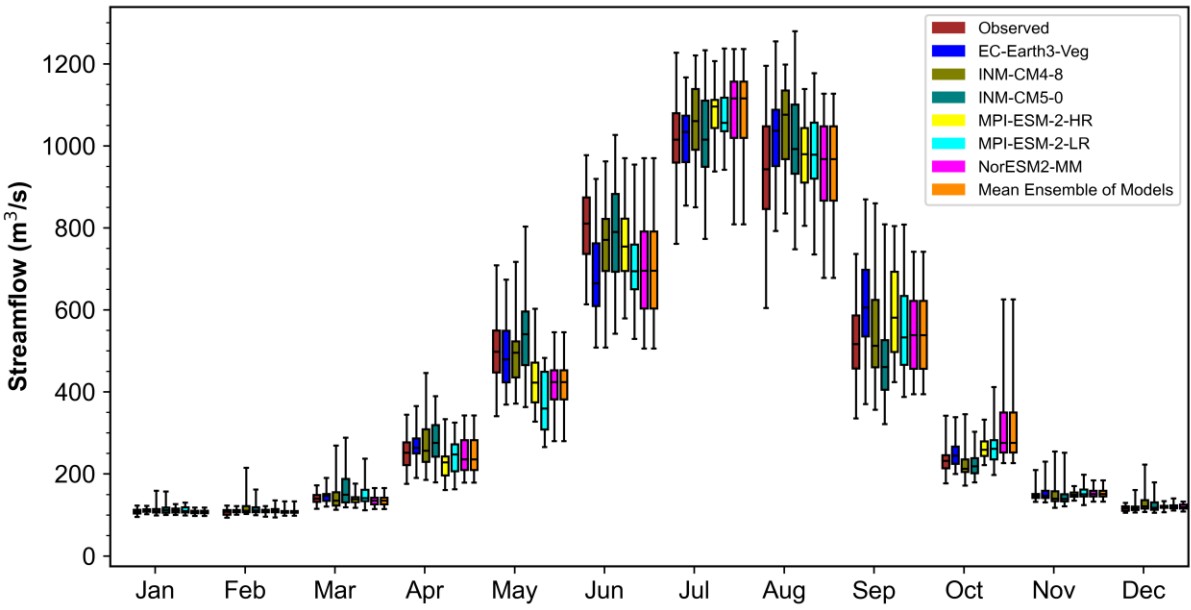


**Figure 9: Box-plot comparing observed and CMIP6-GCMs (mean ensemble of models) simulated streamflow for**
**various months of the year, derived over the period of 1979–2009. The line inside the box denotes the median values**
**of streamflow, while the upper and lower whiskers indicate the highest and minimum values, respectively.**





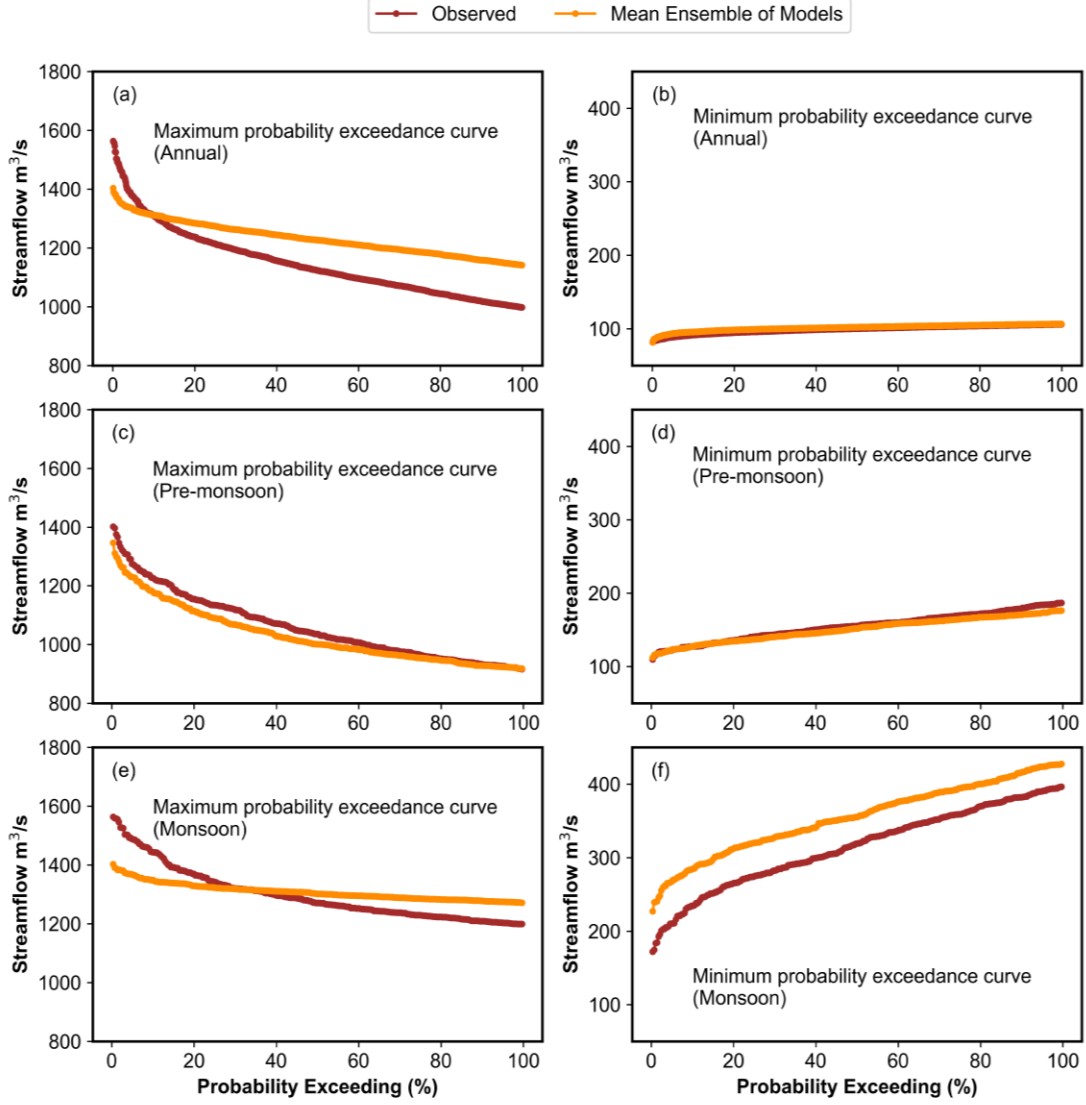


Figure 10: Probability exceedance curves developed using 10% of the highest and lowest flows from the observed and CMIP6-GCMs (mean ensemble of models) over the time span of 1979–2009 for annual and seasonal (pre-monsoon and monsoon) flows.







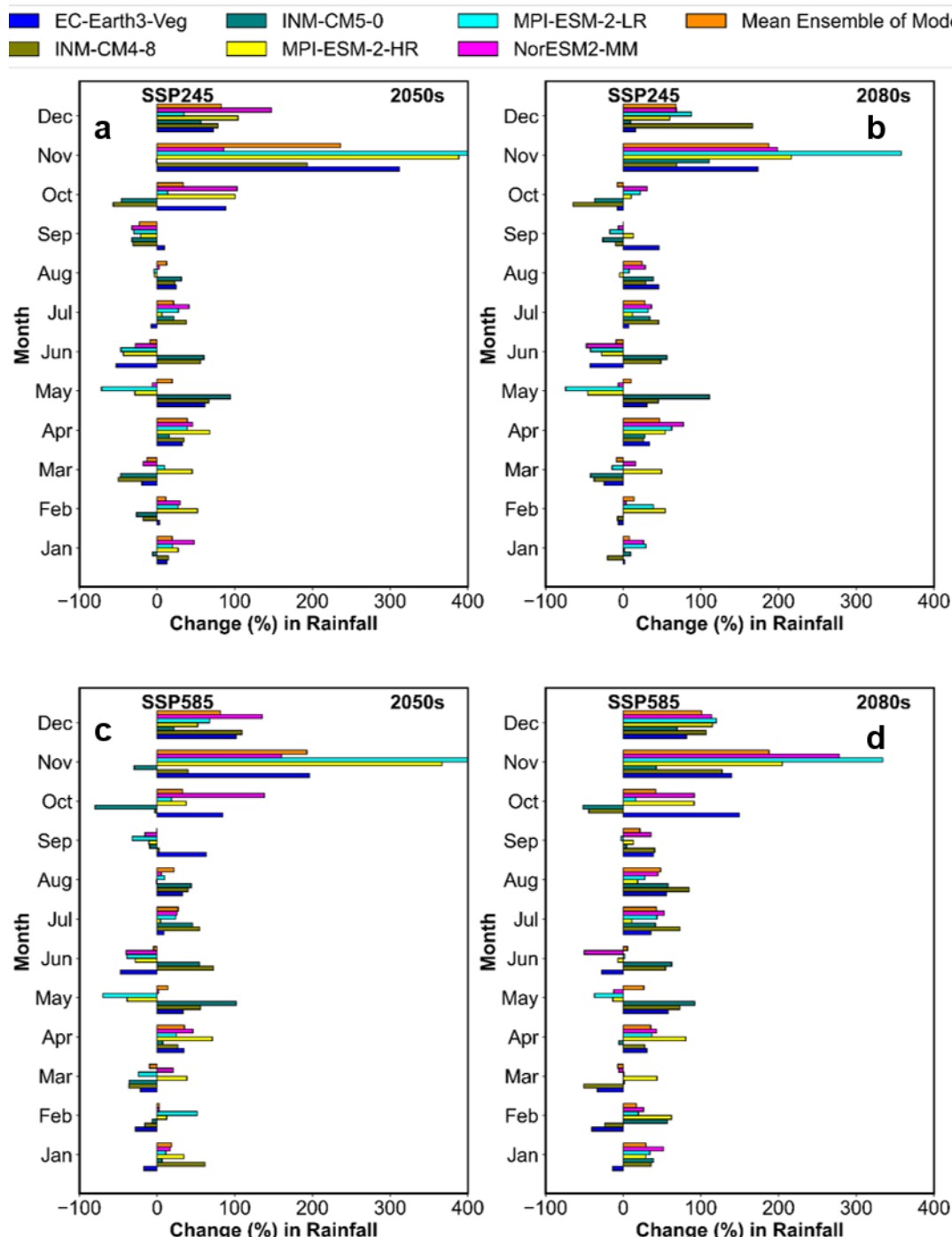


**Figure 11: Projected change in mean monthly rainfall in the sub-basin using different CMIP6-GCMs under SSP245**
**and SSP585 scenarios in the 2050s (Fig.11a and Fig.11b) and 2080s (Fig.11c and Fig.11d).**


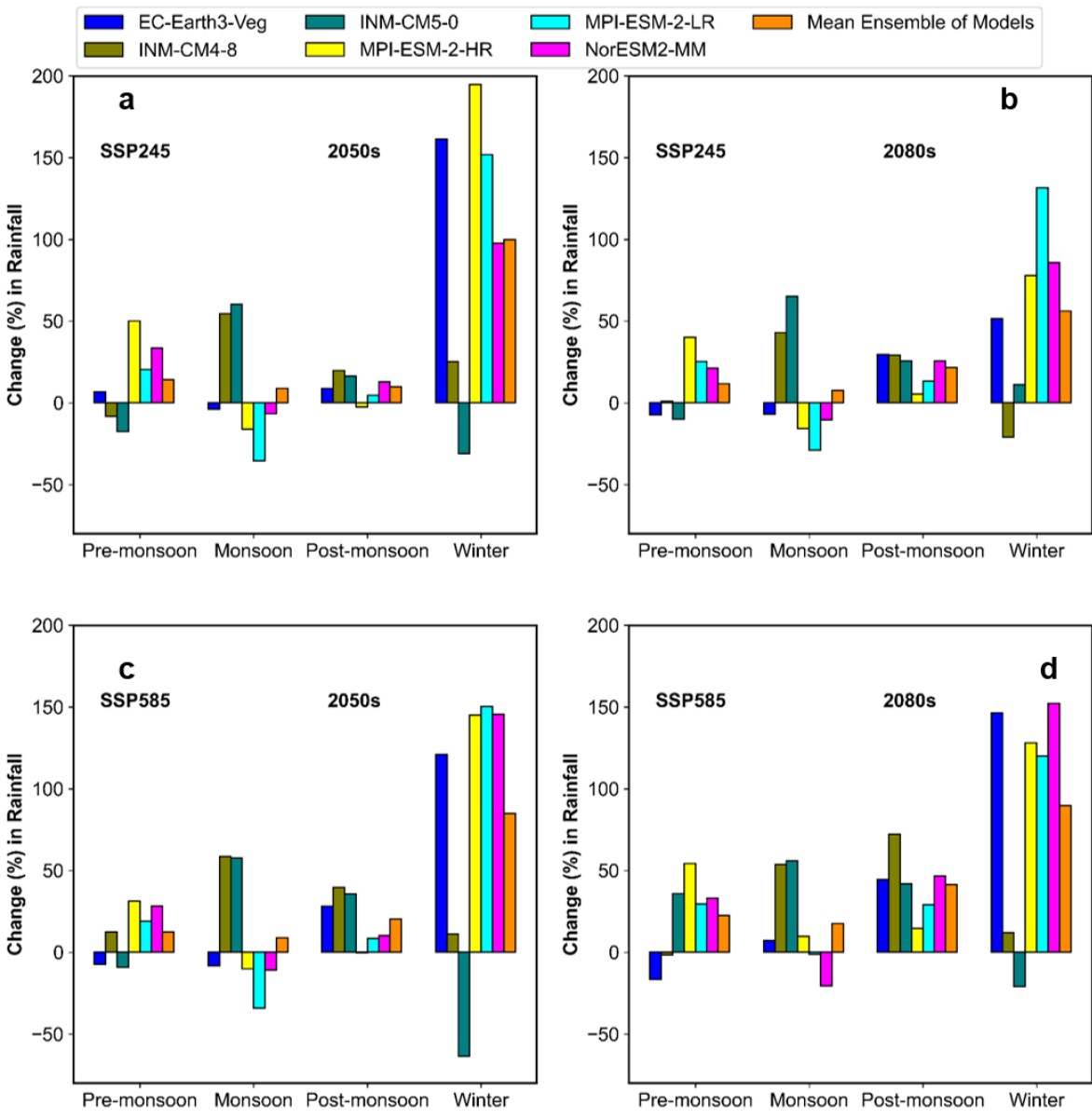


**Figure 12: Projected change in mean seasonal rainfall in the sub-basin using different CMIP6-GCMs under SSP245**
**and SSP585 scenarios in the 2050s (Fig.12a and Fig.12c) and 2080s (Fig.12b and Fig.12d).**


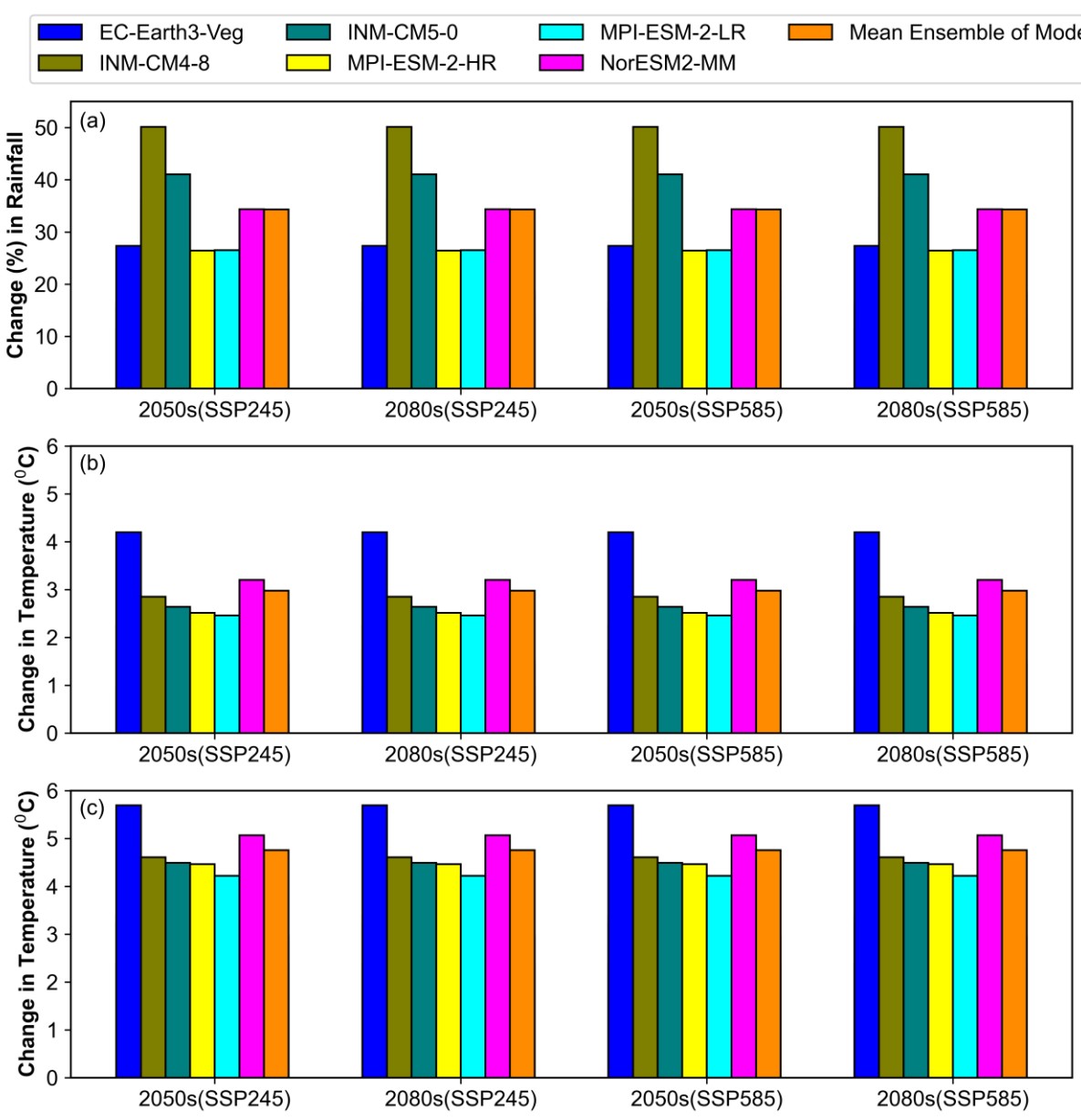


**Figure 13: Projected changes in mean annual rainfall (Fig.13a), T$_{max}$ (Fig.13b) and T$_{min}$ (Fig.13c) in the sub-basin using different CMIP6-GCMs under SSP245 and SSP585 scenarios in the 2050s and 2080s.**




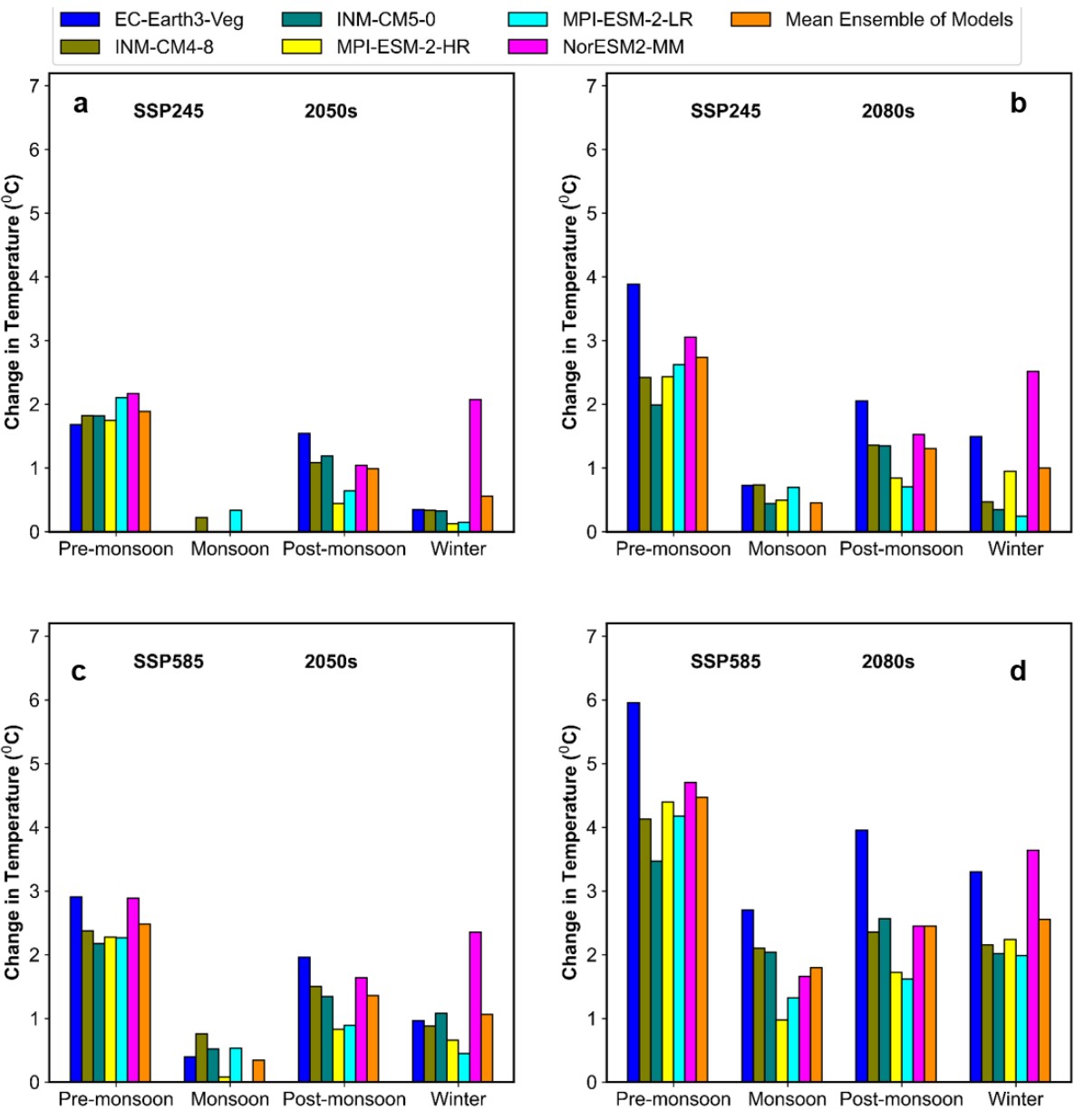


**Figure 14: Projected change in mean seasonal maximum temperature (T_max) in the sub-basin using different CMIP6-**
**GCMs under SSP245 and SSP585 scenarios in the 2050s (Fig.14a and Fig.14c) and 2080s (Fig.14b and Fig.14d).**

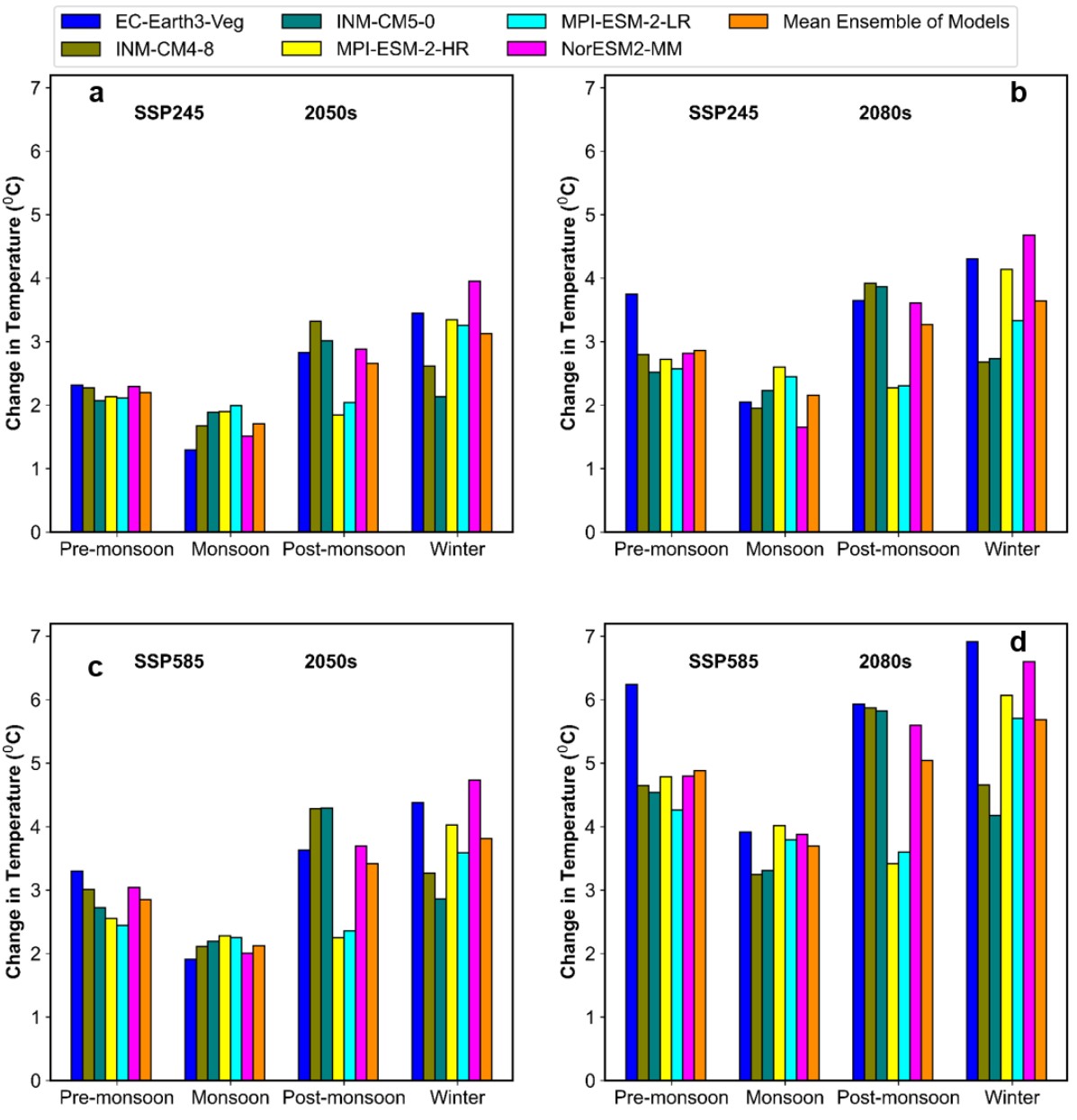


**Figure 15: Projected changes in mean seasonal minimum temperature (T<sub>min</sub>) in the sub-basin using different CMIP6-**
**GCMs under SSP245 and SSP585 scenarios in the 2050s (Fig.15a and Fig.15c) and 2080s (Fig.15b and Fig.15d).**

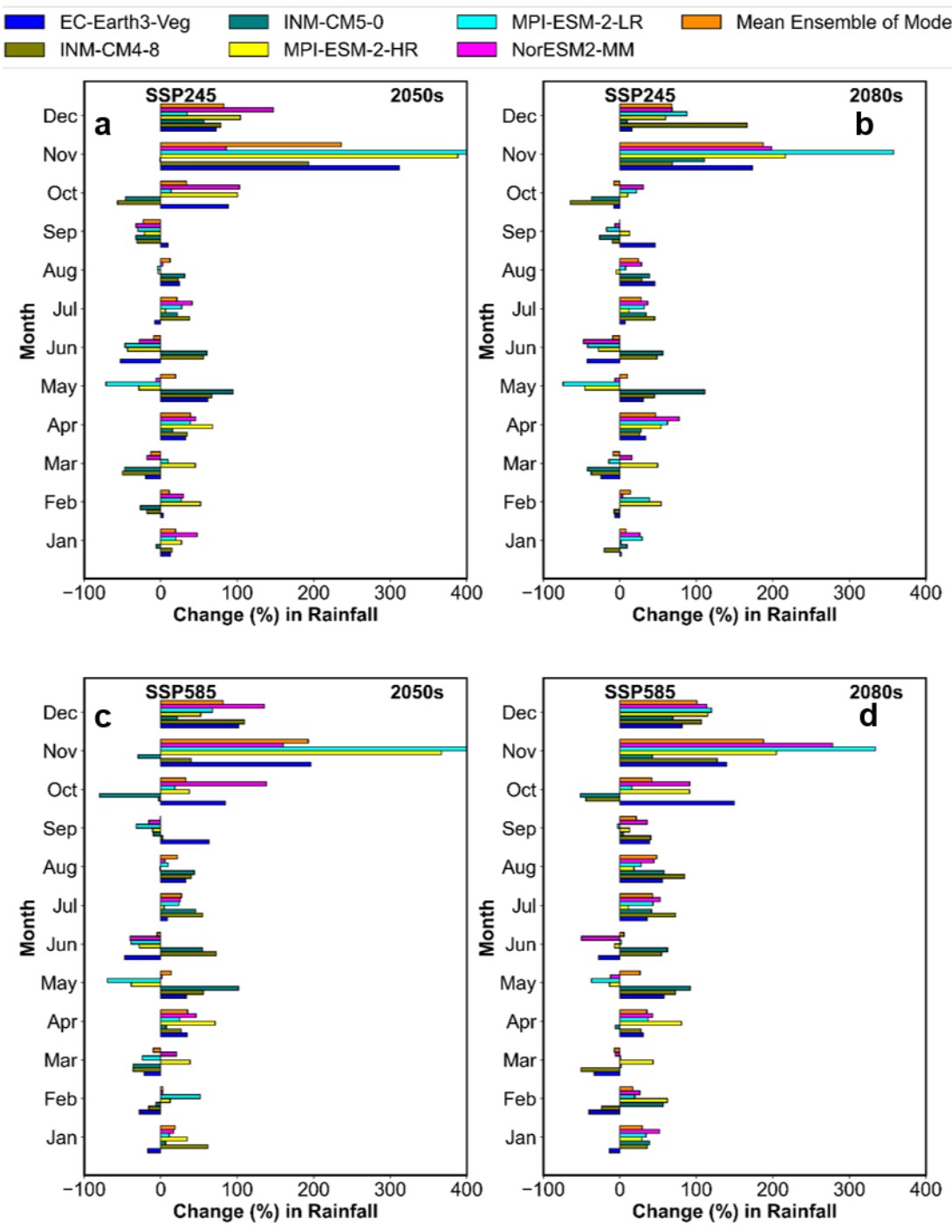


Figure 16: Predicted change in monthly streamflow pattern of the Sutlej River with respect to the reference period
(1979-2009) in 2050s (Fig. 16a and Fig. 16b) and 2080s (Fig. 16c and Fig.16d) under SSP245 and SSP585 for different
CMIP6-GCMs.



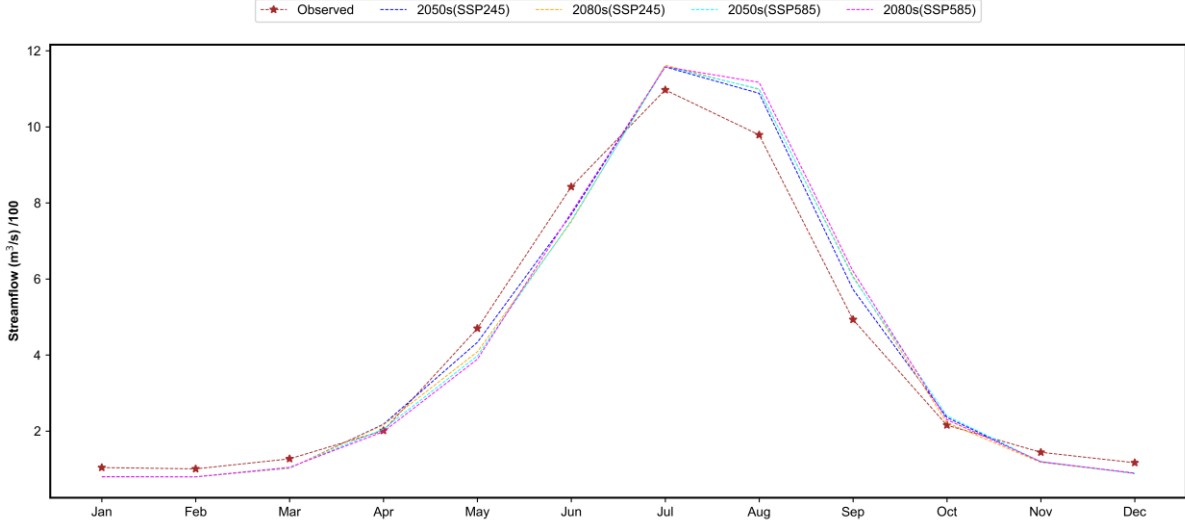


**Figure17: Comparison of monthly observed (1979-2009) and projected discharge of the multi-model ensembles for**
**period 2050s and 2080s under SSP245 and SSP585 scenarios.**

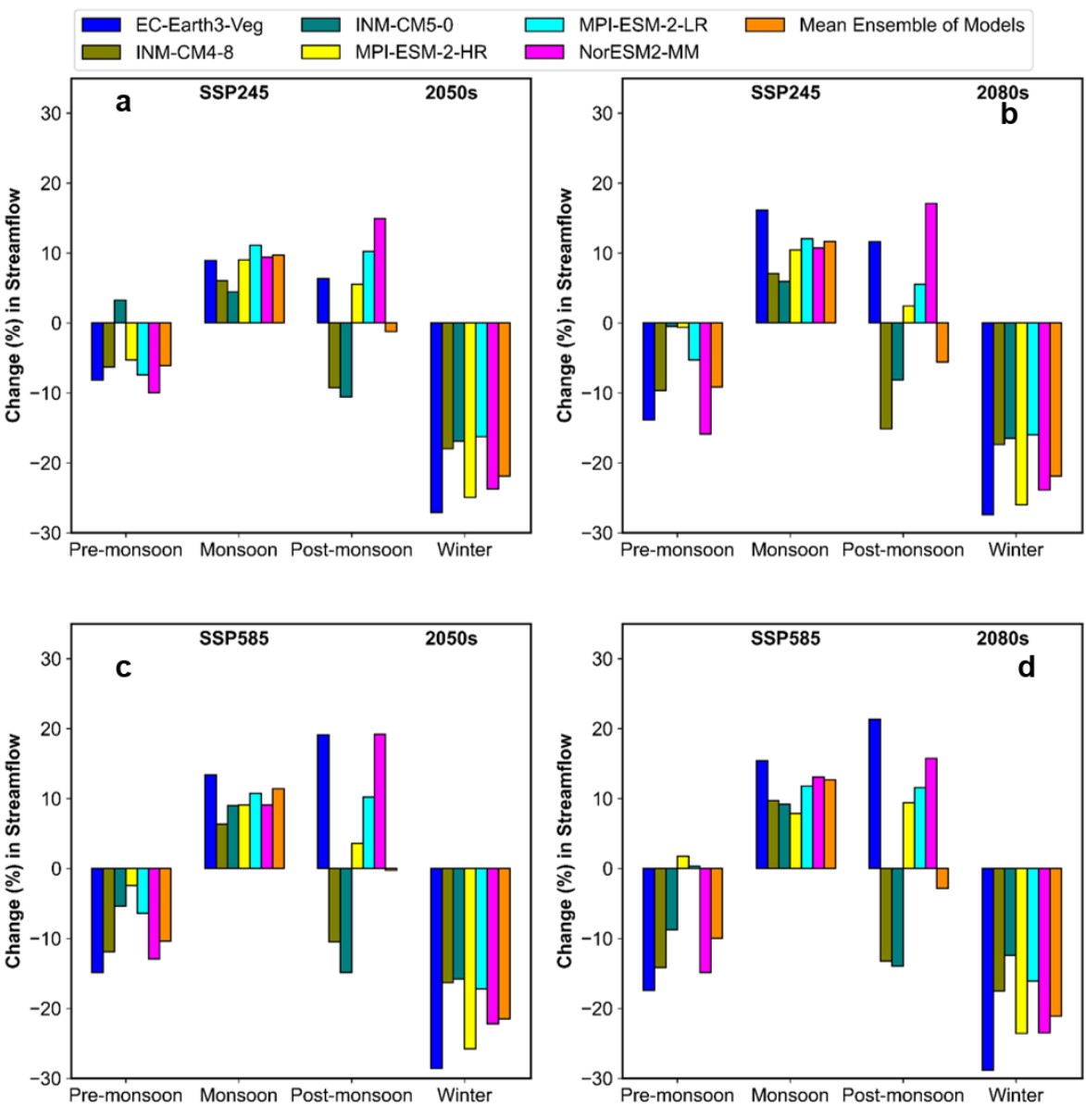

Figure 18: Predicted change in seasonal streamflow pattern of the Sutlej River with respect to the reference period (1979-2009) in 2050s (Fig.18a and Fig.18c) and 2080s (Fig.18c and Fig.18d) under SSP245 and SSP585 for different GCMs.

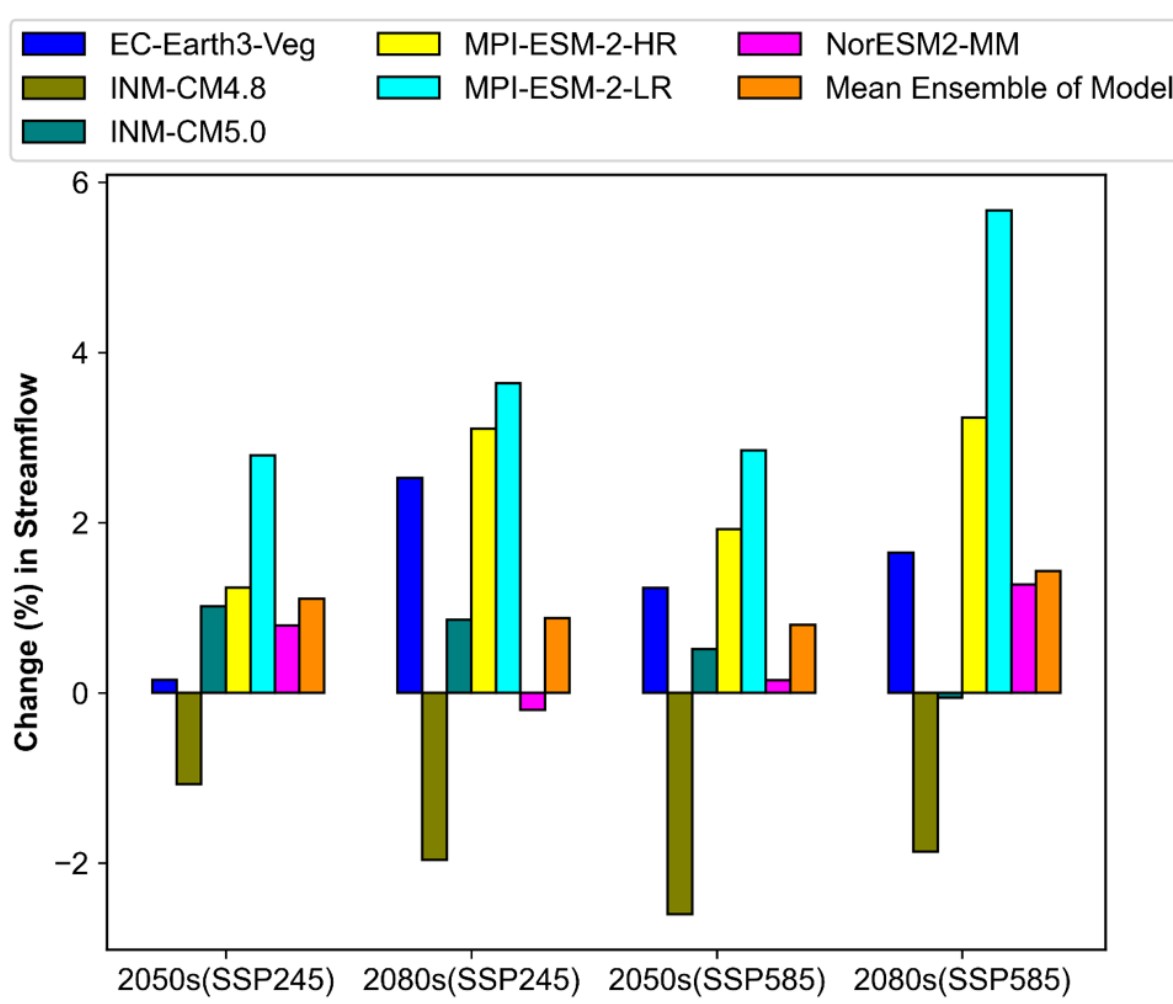


Figure 19: Predicted change in mean annual streamflow of the Sutlej River with respect to the reference period
(1979-2009) in 2050s and 2080s under SSP245 and SSP585 for different GCMs.



**Table 1:** Characteristics of the study catchment over the evaluation period of 1979–2009

| Parameters | Details |
|---|---|
| Details of the sub-catchment | |
| Drainage area of the sub-catchment ($km^2$) | 2457 $km^2$ |
| Altitude | ~500-5000 m |
| Slope | 0-80° |
| Geology | Granite, Jutogh formation and Chail/Salkhala/Hemanta formation |
| Soil | Dystric cambisols, dystric regosols, and eutric fluviosols. |
| Streamflow measured at the outlet (Kasol) of the sub-catchment | |
| Average of annual streamflow | 411.2 $m^3$/s |
| Minimum streamflow (daily) | 64.3$m^3$/s |
| Maximum streamflow (daily) | 2891 $m^3$/s |
| Standard deviation (SD) of annual streamflow | 1750.7$m^3$/s |
| Coefficient of variation (CV) of annual streamflow | 0.1$m^3$/s |
| Rainfall integrated over the sub-catchment | |
| Average of annual rainfall | 1001.3mm |
| Average of monsoon rainfall (July-September) | 403.0mm |
| Average of winter rainfall (December-March) | 277.3mm |
| Temperature integrated over the sub-catchment | |
| Average annual maximum temperature ($T_{max}$) | 28.3°C |
| Average annual minimum temperature ($T_{min}$) | 13.9°C |



**Table 2:** The information on hyper parameters used for estimating model parameters

| Model Name | Hyperparameter | Values |
|---|---|---|
| Artificial Neural Network (ANN) | build_fn, | value = build_regressor |
| | warm_start, | value = False |
| | random_state, | value = None |
| | optimizer, | value = rmsprop |
| | loss, | value = None |
| | metrics, | value = None |
| | batch_size, | value = 64 |
| | validation_batch_size, | value = None |
| | verbose, | value = 1 |
| | callbacks, | value = None |
| | validation_split, | value = 0.0 |
| | shuffle, | value = True |
| | run_eagerly, | value = False |
| | epochs, | value = 500 |
| Generalized Additive Model (GAM) | formula, | value = None |
| | family, | value = gaussian() |
| | data, | value = list() |
| | weights, | value = Null |
| | subset , | value = Null |
| | na.action,offset, | value = Null |
| | method, | value = "GCV.Cp" |
| | optimizer, | value = c("outer","newton") |
| | control, | value = list(), |
| | scale, | value = 0 |
| | select, | value = False |
| | knots, | value = Null |
| | sp, | value = Null |
| | min.sp, | value = Null |
| | H, | value – Null, |
| | gamma, | value = 1 |
| | fit, | value = True |
| | paraPen, | value = Null |
| | G, | value = Null |
| | drop.unused.levels, | value = True |
| | drop.intercept, | value = Null |
| | discrete, | value = False |
| Generalized Linear Model (GLM) | endog, | value = 1D |
| | exog, | value = 1D |

| | | |
|---|---|---|
| | family, | value = sm.families.Gaussian(sm.families.links.log()) |
| | offset, | value = None |
| | exposure, | value = None |
| | freq_weights, | value = None |
| | var_weights, | value = None |
| | missing, | value = str |
| Multivariate Adaptive Regression Splines (MARS) | max_terms, | value = 20 |
| | max_degree , | value = 3 |
| | allow_missing, | value = False |
| | penalty, | value = 3.0 |
| | endspan_alpha, | value = 0.005 |
| | endspan, | value = -1 |
| | minspan_alpha, | value = 0.005 |
| | minspan, | value = -1 |
| | thresh , | value = 0.001 |
| | zero_tol, | value = 1e-12 |
| | min_search_points, | value = 100 |
| | check_every, | value = -1 |
| | allow_linear, | value = True |
| | use_fast, | value = False |
| | fast_K, | value = 5 |
| | fast_h, | value = 1 |
| | smooth, | value = False |
| | enable_pruning, | value = True |
| | feature_importance_type, | value = None |
| | feature_importance_type, | value = 0 |
| Random Forest (RF) | n_estimators, | value=500 |
| | criterion, | value="squared_error" |
| | max_depth, | value=None |
| | min_samples_split, | value = 2 |
| | min_samples_leaf, | value = 5 |
| | min_weight_fraction_leaf, | value = 0.0 |
| | max_features, | value = auto |
| | max_leaf_nodes, | value = None |
| | min_impurity_decrease, | value = 0.0 |
| 1-Dimensional Convolution neural network | Conv1D_filter, | Value = 64 |

| (1D-CNN) | Conv1D_kernel_size, | Value = 2 |
| | Conv1D_pool_size, | Value =2 |
| | Learning rate, | Value = 0.0001 |
| | Epoc, | Value = 30 |
| | Batch size, | Value = 280 |
| | loss | Value = MSE |










