# Peer review of "Machine learning and deep learning based streamflow prediction in a hilly catchment for future scenarios using CMIP6-GCMs data"

_Hydrology and Earth System Sciences, 2022_

## Referee Comment (RC2)

**General Comments**

**Manuscript Title:** Machine learning based streamflow prediction in a hilly catchment for future scenarios 2 using CMIP6 data

**Manuscript No.:** hess-2022-339

The Himalayan river system is most susceptible to the climate change and as for as India is concerned, it vast population depends on the waters of the Himalayan rivers for irrigation, hydropower generation, domestic and other uses. Any change in the water availability (increase or decrease) will definitely impact the downstream population and the ecosystem as a whole. Looking into the fragility of the Himalayan ecosystem, an assessing of the impacts of the climate change on the streamflow using the latest ML techniques such as including the Gaussian Linear Regression Model (GLM), Gaussian 30 Generalized Additive Model (GAM), Multivariate Adaptive Regression Splines (MARS), 31 Artificial Neural Network (ANN), and Random Forest (RF) is the techno-socio need of the hour, particularly in the Himalayas. Six CMIP6 models, two SSP scenarios and four rainfall scenarios (this is really interesting-the lagging concept) for future stream flow predictions at different temporal scale is really interesting and will be immensely helpful to the stakeholders of the region.

This assessment made in this study will be useful in developing water resources development and management plans in the downstream of the basin. The techniques, calibration, validation and length of the records is beyond the question and suffice for such a study. The techniques are perfect and the results are well discussed. I was just flowing through the text and the different sections of the paper. The paper is well written, smooth and the readers will find it amicably understandable. The language is perfect.

Therefore, looking into the applicability and technical enrichments of the manuscript, I will recommend for publication of this manuscript in this journal with minor corrections as given here.

**Specific Comments**

1. The criteria for selection of the GCMs may please be explained at the suitable place in the manuscript.
2. The conclusion part may be written in bullet form for enhanced understanding.

3. A separate section of the future scope of the research will further enrich the need and advancement of such studies.

**Some Typos and minor:**

238 : These were coefficient of determination ($R^2$). The eqn for $R^2$ is missing in the text. ?

248: ……………refers to the standard deviation of observed values. Please correct the STDEVobs?

261: please write the unit of MAE?

457: Thus, the outcomes of the overall study indicate that the RF
* * *

---

## Author Response (AR1)

**Response to the Editor and Reviewer's Comments**

**Editor's comments:** "*Thanks for submitting your interesting manuscript. By now, two reviewers have returned their comments. One of the reviewers raised the serious issue of novelty. I suggest the authors consider the two comments in the following when revising your manuscript: 1) Adding more literatures about machine-learning based studies on climate change in mountainous areas; 2) Besides quite traditional ML models like GLM, GAM, RF, ANN, MARS, the authors should include at least one up-to-date deep learning algorithm for comparison*".

**Response:** We are grateful for the feedback from the Editor and the two anonymous reviewers, whose suggestions helped us greatly to improve the manuscript's overall structure and content. We have added more literature about machine-learning-based studies on climate change in mountainous areas (Lines: 110-171) as well as one deep learning algorithm (1D-CNN) for the comparison (see Method Section 3.2; Lines; 312-325, and Table 2) in the revised paper to incorporate both of the editor's suggestions. These insertions are marked with track-change in the annotated manuscript.

**Reviewer#1**

**General Comments:** "*Thanks to the authors for the efforts in the work and the manuscript. This paper investigated the performance of five machine learning models in streamflow prediction in a sub-catchment in the Sutlej River Basin and assessed the future streamflow change by driving one of the machine learning models with CMIP6 data. The results of this study can give information of future streamflow patterns for this specific region. The presentation is overall satisfactory but some arguments are not scientifically solid enough and requires detailed information. There are some major issues regarding the significance and novelty of the study that I would like the authors to clarify, which are required by the journal of Hydrology and Earth System Sciences. Meanwhile, the structure of the paper needs revision to avoid redundant information. The comments are below:*"

**Reply:** We are thankful to the reviewer for his detailed comprehensive assessment and positive feedback on our manuscript and its potential to generate impact. The manuscript has been revised in accordance with the reviewer's suggestions, and a response to each comment is provided below.

**Comment#1:** "*Regarding the novelty of the paper, the paper argues that very few research has been undertaken for a mountainous catchment, which I do not agree. There are plenty of studies investigating all kinds of machine learning models on streamflow simulation across the world, which covers many mountainous areas, except, they are not marked as mountainous areas specifically. In my opinion, investigating a mountainous area is not a solid argument for the novelty of this paper*".

**Reply:** Yes, we concur with the reviewer that number of studies have used machine learning models to simulate streamflow across the world but these studies are generally limited to observable periods and resulting forecasts. The Himalayan region, one of the most vulnerable region to global climate change, has only received a small number of such research. Process-based hydrological models cannot be used effectively in the area due to the lack of long-term station records; consequently, efficient and successful ML/DL model testing could provide insight into how changing climatic conditions affect the Himalayan river systems. Additionally, none of these investigations across the Sutlej River Basin (SRB) have been conducted thus far using these techniques. Further, an original contribution to knowledge does not only require the use of a new method or technique, but it can also

derive from the integration of new datasets different in nature and origin. Accordingly, in this work, different datasets were used from observations and models, the latter including the sixth and latest phase of the Coupled Model Intercomparison Project (CMIP6) to investigate the potential impact of climate change on the pattern of streamflow in the future. The scenarios from the CMIP6 models are likely to be more realistic than previous generations, i.e., CMIP3 and CMIP5, given their significant improvement in simulating rainfall and temperature for historical records, which are the principal drivers for the runoff generation in the catchment. Therefore, projected changes in seasonal and annual streamflow pattern derived from this study would provide a better insight over the future hydrological regime of the catchment than the previous ones and may assist in devising a better strategy for the operation of hydroelectric power projects and water resources management in the catchment. This sub-basin is bestowed with the large hydropower potential. There are three major hydroelectric power projects: Sunni Dam Project of 1080 MW, Rampur Hydroelectric Power Project (RHEP) of 412 MW, and Nathpa Jhakari Hydro-electric Power Project (NJHEP) of 1500 MW. Besides, Bhakara-Nangal dam (water storage capacity: 9.34 billion cubic meters; Power generation:1500MW) is also located downstream of the river. The projected (mean ensemble) increase in discharge of the river during monsoon season (July-September) may also result in an increased sediment load, which will affect the storage capacity of reservoirs and hydropower potential. However, predicted declines in Sutlej River streamflow over the pre-monsoon (April to June) and winter (December to March) seasons might have a significant impact on agriculture downstream of the river, which is already having problems due to water shortage at this time of year. These sentences were added to the revised manuscript's Introduction (Lines: 145-171) and Discussion (Lines; 589-630) sections to emphasise the work's original contribution to knowledge.

**Comment#2:** *"Regarding the interpretation of the future streamflow patterns, as I understand, the relative change in the paper is to compare the predicted streamflow from CMIP6 data with the observed streamflow in the reference period. Since there are meteorological data in CMIP6 in the reference period, which can be used as inputs for the machine learning models and generates "reference" streamflow data series. With this reference streamflow, the bias of the CMIP6 models to the observations can be excluded. In other words, the relative change in the paper cannot distinguish itself from the bias of CMIP6 models. This will make the results less reliable when the authors argue the results can assist in strategy planning".*

**Reply:** We welcome the reviewer bringing forward this important aspect for discussion. The developed RF (random forest) model have also been trained using historical CMIP6 model projections for simulating streamflow for the reference period (1979-2009). As recommended by the reviewer, we assessed the bias between observed and CMIP6 simulated streamflow (Section 3.4; Lines: 382-395) using following bias correction method (Hawkins et al., 2013):

$$Q_{bc} = \bar{Q}_{obs} + \frac{\sigma_{obs}}{\sigma_{sim}} \left( Q_{future} - \bar{Q}_{sim} \right)$$

where, $Q_{bc}$ and $Q_{future}$ is the bias corrected and raw daily discharge for future simulation, respectively. $\bar{Q}_{obs}$ and $\bar{Q}_{sim)}$ is the mean discharge of observed and historical simulation for reference period, respectively. $\sigma_{obs}$ and $\sigma_{sim}$ is the standard deviation in observed and historical simulation for reference period, respectively. This

method captures variability in both observation and GCMs simulations, which is the interest of this study. Furthermore, the results are also revised accordingly for future streamflow projections (Section 4.2; Lines; 448-478 and Section 4.4; lines:522-585). This has allowed us both to generalise our findings and to make our conclusions comparable to scientific theories, which was previously very challenging to do.

- Hawkins, E., Osborne, T. M., Ho, C. K., & Challinor, A. J. (2013). Calibration and bias correction of climate projections for crop modelling: an idealised case study over Europe. *Agricultural and forest meteorology*, *170*, 19-31.

**Comment#3:** *"The paper investigated only a small subbasin in the Sutlej River Basin (less than 10% in terms of the area), but a lot of description focuses on the whole river basin, which makes it confusing sometimes".*
**Reply:** We incorporated suggestion of the reviewer and removed information that had no direct bearing on the sub-basin (Lines: 198-218).

**Comment#4:** *"Line 35, what is the criteria for selecting these six models?"*
**Reply:** The ranking of GCMs was done to find out the most appropriate models to be used in streamflow projection. Taylor diagram (Taylor, 2001), a robust graphical plot that integrates three statistical metrics, degree of correlation (R), centered root-mean-square error (CRMSE) and ratio of spatial standard deviation (SD) was used to visually analyse the performance of each GCM. Combining these metrics allows determining the degree of pattern correspondence and explaining how exactly a model represents the observed climate. Based on the results of the Taylor diagram, the first six GCMs which showed a good agreement with the observed data (rainfall, $T_{max}$ and $T_{min}$) for a reference period 1979-2009 were selected to examine future patterns in streamflow for the period 2021-2100 in the Sutlej River according to two GHG emissions scenarios: SSP245 and SSP585. This has also been added to the revised manuscript (Lines: 256-289).

**Comment#5:** *"Line 64, are these results from Dai's research also? Please add the reference in a proper way".*
**Reply:** Yes. It has been added in the revised manuscript (Line 63).

**Comment#6:** *"Line 66, what exactly is the word "similar" here referred to? As you mentioned both decreasing/increasing trends in the previous sentence".*
**Reply:** The word "similar" refers that other researchers have also observed both increasing as well decreasing trend in river discharge as Dai et al (2004) reported in their research.

**Comment#7: "***Line 70, please list some examples of other drivers here".*
**Reply:** Precipitation (snowfall + rainfall), temperature, evapotranspiration, snowmelt timing and, snowmelt are the main drivers of runoff generation in a catchment. These are added in the revised manuscript (Lines: 69-71).

*Comment#8: Line 74, what do you mean by adverse effect here?*
**Reply:** The sentence in Line 74 discuss how variability in rainfall pattern may alter hydrological cycle. Thus, the meaning of adverse effect is synonymous to 'alternation' here. To avoid further confusion, we replaced the term 'have an adverse effect on' with 'alter' in the revised manuscript (Line 74)

*Comment#9: Line 84, "generate" should be "generates".*

**Reply:** The suggestion of the reviewer has been incorporated in the revised manuscript (Line 83).

*Comment#10: Line 85, "could" maybe better change to "can".*

**Reply:** In Line 85, "could" has been changed to "can" (Line 84)

*Comment#11: Line 137, the application of ML model should not be the novelty, as ML models are only tools. Consider address this by specifying the scientific questions.*

**Reply:** The suggestion of the reviewer has been well taken and novelty of the present work is explained explicitly in the reply of the comment 1.

*Comment#12: Line 145-150, This is redundant information with Line 122-127.*

**Reply:** As suggested by the reviewer, Line 145-150 has been removed in the revised manuscript to avoid redundancy in the paper.

*Comment#13: Line 151, so the study area is a sub basin of the Sutlej river. Then the description of the whole basin is way too much. Please instead focus on the description of the actual study area.*

**Reply:** The suggestion of the reviewer has been well taken and we shortened the study area description by focusing on the description of the actual study area (Lines; 198-218).

*Comment#14: Line 154, the stations in the figure, are they meteorological stations or hydrological stations?*

**Reply:** These are hydro-meteorological stations, so information on temperature, rainfall, and discharge are all collected at the same location just a few metres apart.

*Comment#15: Line 162. Please check the numbers in the Table, or explain why the mean streamflow is much larger than the maximum flow. And there is no need to give two digits for these variables.*

**Reply:** The number shown in the Table 1 is correct. The streamflow is shown in three categories: mean annual streamflow averaged over the period of 31 years, the highest value of daily maximum streamflow and the lowest value of the daily minimum streamflow during this period. This has now been updated in the Table 1.

*Comment#16: Line 173, the investigation is conducted for the three stations or only the outlet station? And please explain how you connect the CMIP6 data grid to the station point. Have you considered any areal weights?*

**Reply:** The investigation is conducted at the outlet station. We used downscaled and bias-corrected datasets from six GCMs which are available at grid resolution of 0.25 degree×0.25 degree (Mishra et al., 2020). Empirical Quantile Mapping (EQM) approach was used for removing bias in the data. Seven grids of the downscaled CMIP6-GCMs data cover the study area. The temperature (Tmax and Tmin) data were adjusted for topographical bias by separating the study area into a number of homogenous elevation bands spaced by at an interval of 1000m, and applying a temperature laps rate of 6.5°C/1000m within each grid. A Digital Elevation

Model (DEM) of 30 m spatial resolution derived from CartoSat-1 stereo data (www.bhuvan.nrsc.gov.in) was used for this purpose. The values of rainfall and temperature at each grid were then averaged over the catchment using the Thiessen polygon method in order to provide daily rainfall data integrated at the catchment scale for assessing changes in the future climate with respect to the observed period i.e., 1979-2009). These lines are also added in the revised manuscript (Lines: 233-254).

*Comment#17:* *Line 192, reference is absolutely needed here. It is not convincing how you select the models.*
**Reply:** The criteria for selecting GCMs are explained and added in the revised manuscript with proper citation (Lines: 256-289).

*Comment#18:* *Line 208, I do not think this argument is valid here. To be applied to basins with similar geographical characteristics, the models need to be validated across multiple stations. According to the description in the method, I think there is only one station included in this study.*
**Reply:** Yes, we do agree with the reviewer. Therefore, we have expressed only possibility of exploring potential of the developed model in streamflow simulation under similar geographical and climatological environment. For this, it needs to be validated across the different station. Therefore, to avoid confusion, we have remove this sentence in the revised manuscript.

*Comment#19:* *Line 247, there is no in the equation.*
**Reply:** Thanks for pointing out this typo error. Now, it has been corrected.

*Comment#20:* *Line 241, consider to add the formula of R2 also. As in Line 248 you are explaining R2 together with the other two metrics.*
**Reply:** The suggestion of the reviewer has been well taken and it has been incorporated in the revised manuscript (Line 353).

*Comment#21:* *Line 249 to Line 254, references are needed here. Are these standard categories? Also please rewrite in a more organized way.*
**Reply:** Yes, these are the standard categories for evaluating performances of the models. Moreover, as suggested by the reviewer, sentences are redrafted and we added three more metrics namely NSE, KGE and PBIAS to justify the selection of a data driven model. All related references are added in the revised manuscript (Line 338-381).

*Comment#22:* *Line 300, it is also important to consider ensembles, we need to be careful with the "best" model. So maybe be conservative with the conclusions here.*
**Reply:** This conclusion is only based on comparison of five ML models (**GLM, GAM, MARS, ANN, and RF**) and their performances in simulating streamflow under four rainfall scenarios for reference period using observed data. Here, RF has outperformed other models in terms of statistical efficiency therefore referred as the best model. We believe that there is some misinterpretation from the reviewer as these models are firstly trained

with observed data of reference period and then applied for predicting streamflow in future using GCMs outputs (individual or ensembles of models).

**Comment#23:** *Line 305, about the reference period, are you comparing to the observed streamflow? Since there is reference period in CMIP6 also where you can run your model with these data and generated a reference streamflow series. Which method you are using here? And I think this is important to specify in the method.*

**Reply:** We revised the manuscript as per the suggestion of the reviewer. Please refer to the reply of second comment.

**Comment#24:** *Line 321/642, the results here is very confusing as mean ensemble has a much larger relative change than any of the model individually. Could you explain the reason or show annual data series here?*

**Reply:** The reason for higher relative change in mean ensemble streamflow may be attributed to the higher projected change in precipitation in comparison of temperatures of the ensemble time series. However, for individual GCM model, the higher projected relative change in precipitation is followed by relatively higher change in temperature, causing relative lower changes in projected streamflow than the ensemble. However, this issue is resolved in the revised manuscript by applying bias correction over the projected streamflow data.

**Comment#25:** *Line 336, since the magnitude in the change is very different, actually it's not precise to say they are similar tendencies.*

**Reply:** The sentence is revised as per the suggestion of the reviewer.

**Comment#26:** *Line 352 to 375, a huge paragraph here is describing only the numbers, it will be better to put them in a more organized way and add refined information.*

**Reply:** Thanks for the suggestion. We revised the paragraph and discussed results in terms of important outcomes and explained it. In doing so, we also compared these results with previous studies (Section 5; Discussion; Lines: 711-752).

**Comment#27:** *Line 376, please add explanation of pre-monsoon/monsoon/post-monsoon months.*

**Reply:** The suggestion of the reviewer has been well taken and we added explanation for the change in projected streamflow during pre-monsoon/monsoon/post-monsoon months in the revised manuscript (Section 5; Discussion; Lines: 588-629).

**Comment#28:** *Line 419, here the conclusion is different with the information in Figure 8. There, the change in May is sometimes increase.*

**Reply:** Yes, for the month of May, we did not get a clear picture as some models have shown increase in streamflow and others decrease. However, we observed a clear trend of declining streamflow for the mean ensemble of the models that has been validated with the previous published work (Lines: 607-612).

**Comment#29:** *Line 422/658, considering using different line types.*

**Reply:** We revised Figure 9 which now has been renamed as the Figure 17 and used different line types (e.g. continuous and dashed) and colour to highlight distinctions among the lines.

**General Comments: "***The Himalayan river system is most susceptible to the climate change and as for as India is concerned, it vast population depends on the waters of the Himalayan rivers for irrigation, hydropower generation, domestic and other uses. Any change in the water availability (increase or decrease) will definitely impact the downstream population and the ecosystem as a whole. Looking into the fragility of the Himalayan ecosystem, an assessing of the impacts of the climate change on the streamflow using the latest ML techniques such as the Gaussian Linear Regression Model (GLM), Gaussian 30 Generalized Additive Model (GAM), Multivariate Adaptive Regression Splines (MARS), 31 Artificial Neural Network (ANN), and Random Forest (RF) is the techno-socio need of the hour, particularly in the Himalayas. Six CMIP6 models, two SSP scenarios and four rainfall scenarios (this is really interesting-the lagging concept) for future stream flow predictions at different temporal scale is really interesting and will be immensely helpful to the stakeholders of the region. This assessment made in this study will be useful in developing water resources development and management plans in the downstream of the basin. The techniques, calibration, validation and length of the records is beyond the question and suffice for such a study. The techniques are perfect and the results are well discussed. I was just flowing through the text and the different sections of the paper. The paper is well written, smooth and the readers will find it amicably understandable. The language is perfect. Therefore, looking into the applicability and technical enrichments of the manuscript, I will recommend for publication of this manuscript in this journal with minor corrections as given here*".

**Reply:** We are thankful to the reviewer for his detailed comprehensive assessment and positive feedback on our manuscript and its potential to generate impact.

**Specific Comments**

**Comment#1:** "*The criteria for selection of the GCMs may please be explained at the suitable place in the manuscript*".

**Reply:** The ranking of GCMs was done to find out the most appropriate models to be used in streamflow projection. Taylor diagram (Taylor, 2001), a robust graphical plot that integrates three statistical metrics, degree of correlation (R), centered root-mean-square error (CRMSE) and ratio of spatial standard deviation (SD) was used to visually analyse the performance of each GCM. Combining these metrics allows determining the degree of pattern correspondence and explaining how exactly a model represents the observed climate. Based on the results of the Taylor diagram, the first six GCMs which showed a good agreement with the observed data (rainfall, Tmax and Tmin) for a reference period 1979-2009 were selected to examine future patterns in streamflow for the period 2021-2100 in the Sutlej River according to two GHG emissions scenarios: SSP245 and SSP585. This has also been added to the revised manuscript (Lines: 256-289).

**Comment#2:** "*The conclusion part may be written in bullet form for enhanced understanding*".
**Reply:** The suggestion of the reviewer has been incorporated in the revised manuscript (Lines: 630-677).

**Comment#3:** A separate section of the future scope of the research will further enrich the need and advancement of such studies.

**Reply:** We incorporated the suggestion of the reviewer and added the scope of the research in the revised manuscript (Lines: 668-677).

**Some Typos and minor:**

238 : These were coefficient of determination ($R^2$). The eqn for $R^2$ is missing in the text. ?

Reply: We added equation for the $R^2$ in the revised manuscript (Line 353).

248: ……………refers to the standard deviation of observed values. Please correct the STDEVobs?

Reply: This has been corrected in the revised manuscript (Line 361)

261: please write the unit of MAE?

Reply: This has been added in the revised manuscript (Line 403)

457: Thus, the outcomes of the overall study indicate that the RF

Reply: The suggestion of the reviewer has been incorporated in the revised manuscript.

[revised manuscript text omitted]

---

## Editor Decision (ED1)

Hydrol. Earth Syst. Sci. Discuss., referee comment RC1
https://doi.org/10.5194/hess-2022-339-RC1, 2022
**Comment on hess-2022-339**

Anonymous Referee #1
* * *
Referee comment on "Machine learning based streamflow prediction in a hilly catchment for future scenarios using CMIP6 data" by Dharmaveer Singh et al., Hydrol. Earth Syst. Sci. Discuss., https://doi.org/10.5194/hess-2022-339-RC1, 2022
* * *
Review of **Machine learning based streamflow prediction in a hilly catchment for future scenarios using CMIP6 data** for Hydrology and Earth System Sciences

Thanks to the authors for the efforts in the work and the manuscript. This paper investigated the performance of five machine learning models in streamflow prediction in a sub-catchment in the Sutlej River Basin and assessed the future streamflow change by driving one of the machine learning models with CMIP6 data. The results of this study can give information of future streamflow patterns for this specific region. The presentation is overall satisfactory but some arguments are not scientifically solid enough and requires detailed information. There are some major issues regarding the significance and novelty of the study that I would like the authors to clarify, which are required by the journal of Hydrology and Earth System Sciences. Meanwhile, the structure of the paper needs revision to avoid redundant information. The comments are below:

- Regarding the novelty of the paper, the paper argues that very few research have been undertaken for a mountainous catchment, which I do not agree. There are plenty of studies investigating all kinds of machine learning models on streamflow simulation across the world, which covers many mountainous areas, only except, they are not marked as mountainous areas specifically. In my opinion, investigating a mountainous area is not a solid argument for the novelty of this paper.
- Regarding the interpretation of the future streamflow patterns, as I understand, the relative change in the paper is to compare the predicted streamflow from CMIP6 data with the observed streamflow in the reference period. Since there are meteorological data in CMIP6 in the reference period, which can be used as inputs for the machine learning models and generates "reference" streamflow data series. With this reference streamflow, the bias of the CMIP6 models to the observations can be excluded. In other words, the relative change in the paper cannot distinguish itself from the bias of CMIP6

models. This will make the results less reliable when the authors argue the results can assist in strategy planning.

- The paper investigated only a small subbasin in the Sutlej River Basin (less than 10% in terms of the area), but a lot of description focuses on the whole river basin, which makes it confusing sometimes.

Line by Line comments are below:

Line 35, what is the criteria for selecting these six models?

Line 64, are these results from Dai's research also? Please add the reference in a proper way.

Line 66, what exactly is the word "similar" here referred to? As you mentioned both decreasing/increasing trends in the previous sentence.

Line 70, please list some examples of other drivers here.

Line 74, what do you mean by adverse effect here?

Line 84, "generate" should be "generates".

Line 85, "could" maybe better change to "can".

Line 137, the application of ML model should not be the novelty, as ML models are only tools. Consider address this by specifying the scientific questions.

Line 145-150, This is redundant information with Line 122-127.

Line 151, so the study area is a sub basin of the Sutlej river. Then the description of the

whole basin is way too much. Please instead focus on the description of the actual study area.

Line 154, the stations in the figure, are they meteorological stations or hydrological stations?

Line 162. Please check the numbers in the Table, or explain why the mean streamflow is much larger than the maximum flow. And there is no need to give two digits for these variables.

Line 173, the investigation is conducted for the three stations or only the outlet station? And please explain how you connect the CMIP6 data grid to the station point. Have you considered any areal weights?

Line 192, reference is absolutely needed here. It is not convincing how you select the models.

Line 208, I do not think this argument is valid here. To be applied to basins with similar geographical characteristics, the models need to be validated across multiple stations. According to the description in the method, I think there is only one station included in this study.

Line 247, there is no in the equation.

Line 241, consider to add the formula of $R^2$ also. As in Line 248 you are explaining $R^2$ together with the other two metrics.

Line 249 to Line 254, references are needed here. Are these standard categories? Also please rewrite in a more organized way.

Line 300, it is also important to consider ensembles, we need to be careful with the "best" model. So maybe be conservative with the conclusions here.

Line 305, about the reference period, are you comparing to the observed streamflow? Since there is reference period in CMIP6 also where you can run your model with these data and generated a reference streamflow series. Which method you are using here? And

I think this is important to specify in the method.

Line 321/642, the results here is very confusing as mean ensemble has a much larger relative change than any of the model individually. Could you explain the reason or show annual data series here?

Line 336, since the magnitude in the change is very different, actually it's not precise to say they are similar tendencies.

Line 352 to 375, a huge paragraph here is describing only the numbers, it will be better to put them in a more organized way and add refined information.

Line 376, please add explanation of pre-monsoon/monsoon/post-monsoon months.

Line 419, here the conclusion is different with the information in Figure 8. There, the change in May is sometimes increase.

Line 422/658, considering using different line types.

Machine learning based streamflow prediction in a hilly catchment for future scenarios using CMIP6 data

**General Comments**

**Manuscript Title:** Machine learning based streamflow prediction in a hilly catchment for future scenarios 2 using CMIP6 data

**Manuscript No.:** hess-2022-339

The Himalayan river system is most susceptible to the climate change and as for as India is concerned, it vast population depends on the waters of the Himalayan rivers for irrigation, hydropower generation, domestic and other uses. Any change in the water availability (increase or decrease) will definitely impact the downstream population and the ecosystem as a whole. Looking into the fragility of the Himalayan ecosystem, an assessing of the impacts of the climate change on the streamflow using the latest ML techniques such as including the Gaussian Linear Regression Model (GLM), Gaussian 30 Generalized Additive Model (GAM), Multivariate Adaptive Regression Splines (MARS), 31 Artificial Neural Network (ANN), and Random Forest (RF) is the techno-socio need of the hour, particularly in the Himalayas. Six CMIP6 models, two SSP scenarios and four rainfall scenarios (this is really interesting-the lagging concept) for future stream flow predictions at different temporal scale is really interesting and will be immensely helpful to the stakeholders of the region.

This assessment made in this study will be useful in developing water resources development and management plans in the downstream of the basin. The techniques, calibration, validation and length of the records is beyond the question and suffice for such a study. The techniques are perfect and the results are well discussed. I was just flowing through the text and the different sections of the paper. The paper is well written, smooth and the readers will find it amicably understandable. The language is perfect.

Therefore, looking into the applicability and technical enrichments of the manuscript, I will recommend for publication of this manuscript in this journal with minor corrections as given here.

**Specific Comments**

1. The criteria for selection of the GCMs may please be explained at the suitable place in the manuscript.
2. The conclusion part may be written in bullet form for enhanced understanding.

3. A separate section of the future scope of the research will further enrich the need and advancement of such studies.

**Some Typos and minor:**

238 : These were coefficient of determination ($R^2$). The eqn for $R^2$ is missing in the text. ?

248: ……………refers to the standard deviation of observed values. Please correct the $STDEV_{obs}$?

261: please write the unit of MAE?

457: Thus, the outcomes of the overall study indicate that the RF
* * *